# 3D printed biomimetic cochleae and machine learning co-modelling provides clinical informatics for cochlear implant patients

Iek Man Lei[1,2], Chen Jiang[1,3,4], Chon Lok Lei [5,6], Simone Rosalie de Rijk [3], Yu Chuen Tam [7], Chloe Swords [8], Michael P. F. Sutcliffe[1], George G. Malliaras [1], Manohar Bance[3,9 ✉] & Yan Yan Shery Huang [1,2,9 ✉]

Cochlear implants restore hearing in patients with severe to profound deafness by delivering electrical stimuli inside the cochlea. Understanding stimulus current spread, and how it correlates to patient-dependent factors, is hampered by the poor accessibility of the inner ear and by the lack of clinically-relevant in vitro, in vivo or in silico models. Here, we present 3D printing-neural network co-modelling for interpreting electric field imaging profiles of cochlear implant patients. With tuneable electro-anatomy, the 3D printed cochleae can replicate clinical scenarios of electric field imaging profiles at the off-stimuli positions. The co-modelling framework demonstrated autonomous and robust predictions of patient profiles or cochlear geometry, unfolded the electro-anatomical factors causing current spread, assisted on-demand printing for implant testing, and inferred patients' in vivo cochlear tissue resistivity (estimated mean = 6.6 kΩcm). We anticipate our framework will facilitate physical modelling and digital twin innovations for neuromodulation implants.

[1] Department of Engineering, University of Cambridge, Cambridge, United Kingdom. [2] The Nanoscience Centre, University of Cambridge, Cambridge, United Kingdom. [3] Department of Clinical Neurosciences, University of Cambridge, Cambridge, United Kingdom. [4] Department of Electronic Engineering, Tsinghua University, Beijing 100084, China. [5] Institute of Translational Medicine, Faculty of Health Sciences, University of Macau, Taipa, Macau. [6] Department of Computer Science, University of Oxford, Oxford, United Kingdom. [7] Emmeline Centre for Hearing Implants, Addenbrookes Hospital, Cambridge, United Kingdom. [8] Department of Physiology, Development and Neurosciences, Cambridge, United Kingdom. [9] These authors jointly supervised this work: Manohar Bance, Yan Yan Shery Huang. ✉email: mlb59@cam.ac.uk; yysh2@cam.ac.uk

The use of neuromodulation implants and bioelectronic devices has been increasing rapidly and is anticipated to form a new era of medicine[1,2]. By delivering local electrical stimuli to tissues, these electronic implants restore lost neural functions in tissues or nerves or modulate signalling patterns for therapeutic outcomes[2,3]. Cochlear implants (CIs) are by far the most widely used neuromodulation electronic implants, with well over 500,000 CIs having been implanted worldwide[4], and their prevalence is only expected to grow more rapidly with the projected increase in the elderly population[1,4]. Bypassing the malfunctioning peripheral auditory mechanisms by direct neural stimulation, the CI electrode array is designed to restore sound perception. It also attempts, in broad terms, to reproduce the tonotopic architecture of the cochlea by delivering frequency-specific programmed stimulation at localised regions of the cochlear lumen; this, in turn, stimulates separate auditory neural elements[5,6] (Fig. 1a), with lower sound frequencies represented apically and higher frequencies basally.

A major limitation of today's neural prostheses is their imprecise control of the administered stimulus, arising from the intrinsic conductive nature of biological tissues[7,8] and particularly of the biological fluids in the inner ear[5,9]. This limitation is well exemplified by the 'current spread' problem of CIs, where the uncontrolled spread of electrical stimulus leads to off-target excitation of the neighbouring auditory nerve fibres (thus causing a mismatch or 'smeared' representation in the perceived sound from that intended)[9] (Fig. 1a). Cochlear anatomy, tissue conductivity and implant positioning are suggested to be the primary patient-specific factors controlling the intracochlear voltage distribution induced by CIs[9–12]. In particular, cochlear anatomy (in terms of size and shape) is variable[13], with different levels of volumetric conductance of cochlear fluids affecting the intracochlear voltage induced by stimulation. Moreover, pathophysiological conditions could affect the electrical conductivity of the cochlear bony walls, and thus CI induced electric fields[14]. As the cochlea is embedded deep inside the temporal bone and has a complex anatomy, its electrical characteristics are difficult to quantify in a living subject. As a result, a model that deciphers how different characteristics of a patient's cochlea affect the stimulus spread would be a valuable tool for predicting and optimising the stimulus signals and provide insights into factors controlling the large variation in patient-specific CI performance and sound perception.

Although various physical and computational models have been developed for CI testing[9,12,15–17], they are insufficient to evaluate the stimulus spread in human cochleae. Animal models are well-established for in vivo CI testing, but due to the drastic differences between the cochlear anatomies of humans and animals[18], incomplete insights into human responses are obtained[1,3]. Though human cadavers can provide anatomical fidelity, they are limited in supply and have altered electrical properties due to preservation and post-mortem changes[19]. In silico approaches, such as finite element modelling (FEM), can overcome ethical, sample availability and cost issues[20]. However,

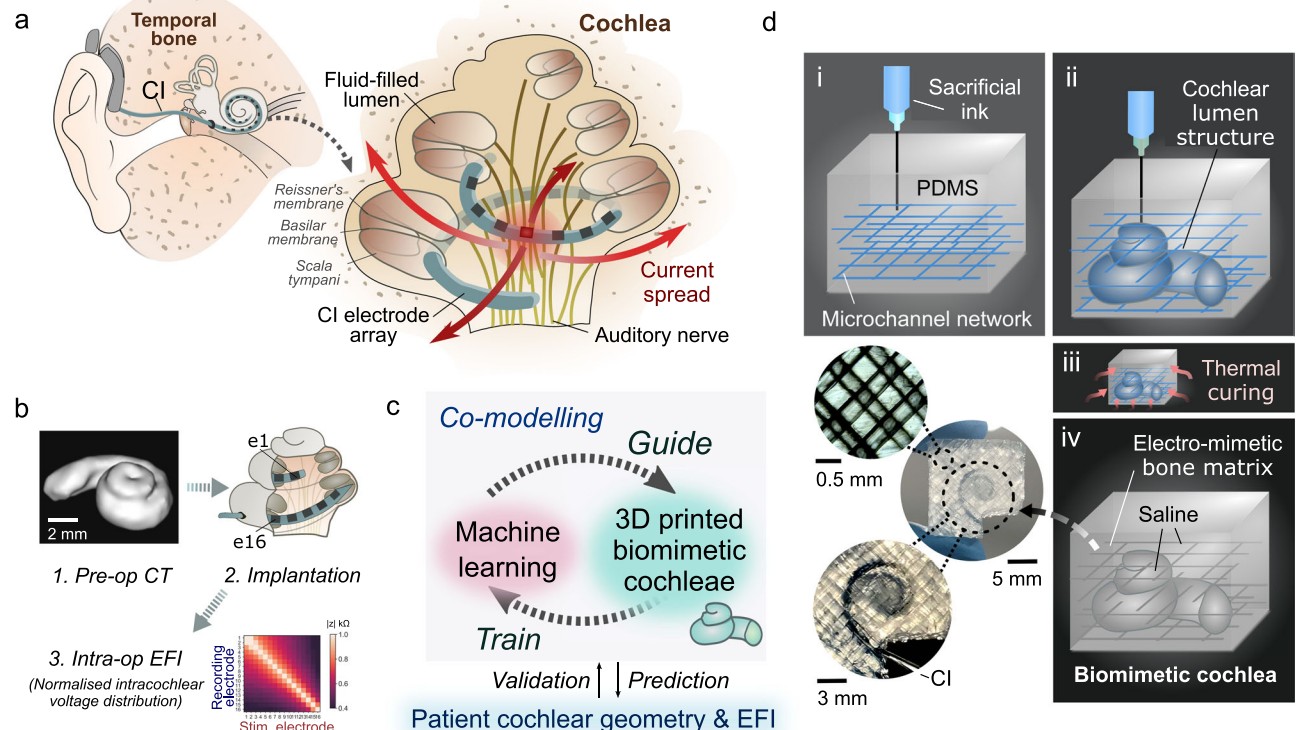

**Fig. 1 3PNN co-modelling approach with embedded 3D printing of biomimetic cochleae for reproducing the CI stimulus spread characteristics. a** Schematic of the auditory system and the cochlea with a CI implanted. The 'current spread' problem induced by a stimulating electrode of the CI electrode array is indicated. **b** Schematic of the routine CI assessment process; **1**. Preoperative CT scan of a patient's cochlea, which typically only has sufficient resolution to reveal the ensemble spiral-shaped cavity of a cochlea; **2**. Implantation of the electrode array of a CI in the scala tympani of the cochlea; **3**. Acquisition of an intra-operative EFI (electric field imaging) profile from a patient, which is derived from recording the induced intracochlear voltage $V$ measured at each electrode upon injecting consecutive current pulses at each electrode in the array. The voltage measurements are then converted to transimpedance magnitude $|z|$ by normalising the voltage $V$ with the stimulation current impulse $I_{stim}$ ($|z| = V/I_{stim}$). The off-stimulation (off-diagonal) measurements in the EFI present information about the tissue impedance[9]. **c** Overview of the 3PNN co-modelling framework for providing clinical informatics. **d** Schematic of the embedded 3D printing strategy to produce the electro-mimetic bone matrices and the biomimetic cochleae.

existing FEM modelling is limited by several factors, including scant knowledge of the electrical properties of live human cochlear tissues to fit different in vivo cases[21], the inability to capture patient-dependent anatomically-guided CI positioning and the underdetermined boundary conditions and physical/empirical law descriptions[15] (discussed in Supplementary Fig. 1a).

Here, we establish a library of 3D printed cochlear models ($n = 82$) for robust modelling of clinical CI testing data. These 3D printed cochleae capture the diverse geometries that human cochlear lumens can take, along with a spectrum of bone tissue resistivities, using ranges reported in in vivo human studies. Supplementary Video 1 shows CT scans of exemplar 3D printed biomimetic cochleae. Using these models, a broad spectrum of clinically representative electric field imaging (EFI) profiles (normalised intracochlear voltage distribution along the CI electrode array) is acquired. Then, by inputting EFI profiles acquired from the biomimetic cochleae as the training dataset, we establish a neural network machine learning model termed 3PNN (3D printing and neural network co-modelling, overview shown in Fig. 1c), which provides powerful clinical informatics such as deciphering patient-specific attributes of CI current spread and inferring patient-dependent cochlear tissue resistivity.

## Results

**Designable electro-mimetic bone matrices.** The human cochlea is a spiral-shaped hollow organ embedded in the temporal bone (Fig. 1a). Since there were no established reports of in vivo cochlear tissue conductivities, our first goal was to establish a printable material system that could emulate the range of reported bone tissue conductivities (hereafter, termed electro-mimetic bone matrix). In vivo human studies estimated that the electrical resistivities of human skulls vary widely between 0.6 to 26.6 kΩcm, depending on the site, composition, age and porosity[22–26] (Supplementary Fig. 2).

To reproduce the mesoscale electrical properties of bone, we take inspiration from the micro-architecture of bones, which consists of conductive fluid-filled interconnected pores surrounded by a poorly conductive mineralised phase[27]. Thus, we structured an electro-mimetic bone matrix that exhibits interconnected saline-filled channels inside a crosslinked PDMS (polydimethylsiloxane) elastomer. The interconnected channels were created by embedded printing a Pluronic F127 sacrificial ink in pre-crosslinked PDMS (Fig. 1d and Supplementary Video 2), permitting flexible and precise tuning of the void density and, therefore, the resistivity of the electro-mimetic bone matrices (Supplementary Fig. 3e). Comparing our printing method with stereolithography, Pluronic F127 can be easily removed after printing[28] and further enhances the wettability of PDMS due to its amphiphilic nature. The channels were then filled with physiological saline, which we hypothesise is important to emulate the electrical impedance properties of bone tissues, as pores in bone are normally wet with extracellular fluids. The electrochemical impedance spectroscopy (EIS) measurements in Fig. 2a and Supplementary Fig. 4 show that an electro-mimetic bone matrix can be designed to exhibit impedance properties matching those of a cadaveric cochlear bone in a human head for the entire frequency range ($f = 10$ Hz–100 kHz) studied in EIS. In particular, the Fourier fundamental frequency associated with the EFI stimulation pulse, (estimated to be $f \sim 14$ to $\sim 20$ kHz depending on CI type), lies in the frequency-independent impedance magnitude plateau region. By varying the void fraction in the electro-mimetic bone matrix from 20 to 84%, the resistivity of the matrix that is derived from the impedance magnitude plateau can be tuned from 0.2 to 23.4 kΩcm (Fig. 2b), covering almost the entire reported resistivity range of live human skull tissues[22–26] (0.6–26.6 kΩcm, Fig. 2c).

Figure 3 shows a material property chart summarising the electrical resistivity and Young's modulus for a range of biological tissues and polymeric materials. The 3D printed electro-mimetic

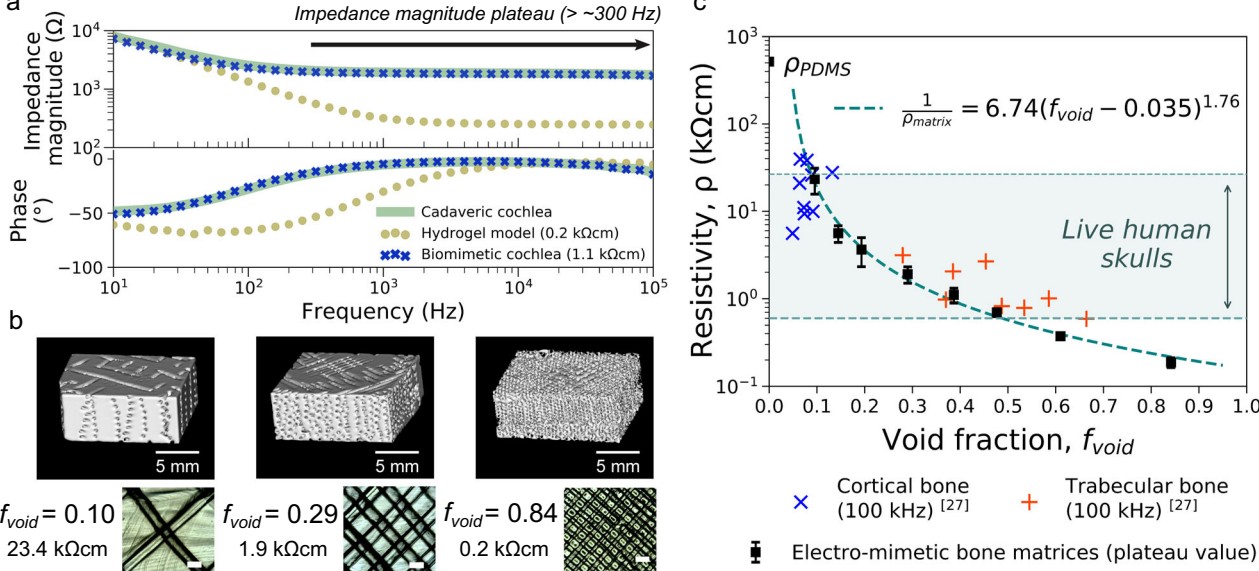

**Fig. 2 Electrical properties of electro-mimetic bone matrices. a** Bode plot showing the impedance properties of a cadaveric cochlea in a human head, and 3D printed cochlear models made of an electro-mimetic bone matrix and a hydrogel. The frequency range associated with the impedance magnitude plateau is indicated. **b** μ-CT reconstructed images (top) and optical microscopic images (bottom) of the electro-mimetic bone matrices at different volumetric void fractions ($f_{void}$). Scale bar of the optical microscopic images = 500 μm. The resistivities of the matrices were determined from their plateau impedance magnitude and the size of the samples. $n = 3$ independent samples. **c** Resistivity of the electro-mimetic bone matrices (plateau value, $n = 3$ independent samples) as a function of $f_{void}$, compared to the reported resistivities of bovine cortical and trabecular bones[27]. The relationship between the resistivity of the electro-mimetic bone matrix and $f_{void}$ is well-described by a percolation equation of a conductor-insulator composite[54] (Supplementary Fig. 3e). Data were presented as mean values ± SD.

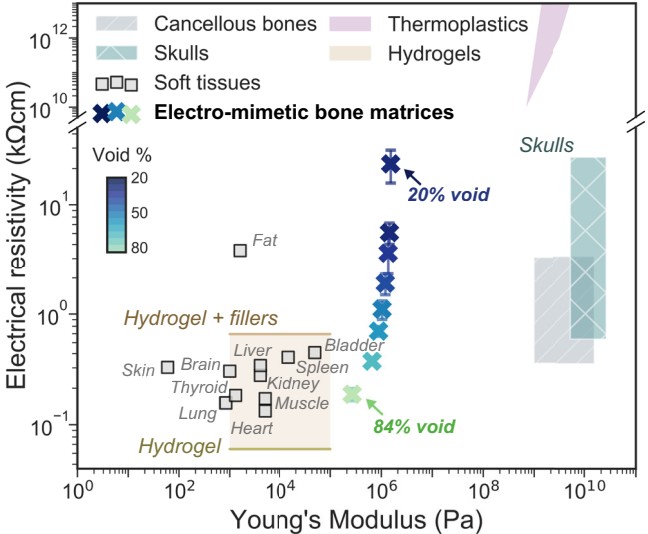

**Fig. 3 Wide resistivity tuneability and adequate mechanical properties of electro-mimetic bone matrices.** A map of resistivity versus Young's modulus of human tissues, thermoplastics, the hydrogel-fillers matrices and the electro-mimetic bone matrices (plateau values) was tested in this study ($n = 3$ independent samples). The compositions of the hydrogel and hydrogel-fillers matrices tested here are listed in Supplementary Fig. 5a. Young's modulus of the electro-mimetic bone matrix was estimated by scaling Young's modulus of pure PDMS (1.7 MPa at a curing temperature of 60 °C[55]) linearly with the $f_{void}$ of the matrix. Tissues and thermoplastics data and Young's modulus of hydrogels were compiled from literature[22-27,56-60]. Data of the electro-mimetic bone matrices are presented as mean values ± SD.

bone matrices cover a wide resistivity range, which cannot be imitated by a single printable material (i.e. thermoplastics or hydrogels) alone or a hydrogel-fillers matrix (i.e. bioceramics and PDMS microbeads dispersed in hydrogels) (Supplementary Fig. 5a). Apart from electrical resistivity, we suggest that Young's modulus of the model is also an important consideration for electronic implant testing. Adopting PDMS as the solid phase of the electro-mimetic bone matrix not only facilitates the ease of embedded printing, but also imparts favourable mechanical properties as a CI testing platform. With Young's modulus in the $10^6$ Pa range, we estimate that the force associated with CI electrode insertion will not induce a significant deformation to the matrix (Supplementary Note 1). At the same time, the compliance of the matrix mitigates mechanical damage to the fine electrodes of a CI, which is commonly experienced when inserting CI electrode arrays repeatedly in cadaveric samples (modulus of hard tissues $>10^9$ Pa). Hence, multiple insertions can take place for the same CI electrode array, which is of practical importance due to the time-consuming fabrication and costs associated with a fully functioning CI. Overall, the above results suggest the electro-mimetic bone matrices be a suitable material system for creating electroanatomical models of human cochleae.

**3D printed biomimetic cochleae.** Clinically, a CI electrode array is inserted into the scala tympani, one of the three cochlear ducts[14] (Fig. 1a). As a coarse-grained approach to replicate the electroanatomical features of a CI implanted cochlea, we approximate the cochlea as one ensemble spiral cavity with continuously narrowing diameter and omit the inner soft-tissue membranous structures inside the cochlea, such as the basilar membrane and Reissner's membrane. This is because, firstly, in a

typical patient's preoperative CT scan as routine clinical assessment (Fig. 1b), the scan resolution only permits the identification of the shape of the ensemble cochlear lumen and not the fine microanatomical soft-tissue structures (Supplementary Fig. 6); and secondly, our preliminary finite element modelling shows that the effect of the basilar membrane and the Reissner's membrane inside a cochlea on the off-stimulation EFI profile is likely to be insignificant, as the boundary impedances are dominated by surrounding bone tissues (see Supplementary Fig. 1b). Therefore, we constructed the biomimetic cochleae by embedded 3D printing a tapered and spiral-shaped cochlear lumen cavity inside an electro-mimetic bone matrix (Fig. 1d and Supplementary Video 2). The spiral-shaped cavity was filled with physiological saline to mimic the ionic conduction milieu in the cochlea (perilymph) (Supplementary Fig. 3c) and the conduction properties at the electrode-electrolyte interface.

Since the size and the shape of a cochlea is unique to each individual and can vary greatly from person-to-person[13,29,30], we assign four geometrical descriptors to parametrically describe the reported anatomical variations in CI implanted human cochleae; they are basal lumen diameter, taper ratio, cochlear width and cochlear height (see definitions in Fig. 4a and Supplementary Table 1). For electroanatomical modelling of cochleae, we incorporated a fifth descriptor, the matrix resistivity, which is controlled by the void fraction of the electro-mimetic bone matrix. In total, 82 biomimetic cochleae were printed at different combinations of model descriptors. With this physical model library, we artificially reconstructed a broad spectrum of the electroanatomical features of human cochleae with even feature distributions.

Figure 4b shows high-resolution μ-CT scans of a cadaveric cochlea and an exemplar 3D printed biomimetic cochlea with a CI inserted. It is worth noting that the CI electrode-to-spiral centre distance displayed in the 3D printed cochleae matches closely with the electrode-to-modiolus distances measured clinically from patients' CT scans[31] (Fig. 4c(i)). Despite only four geometric descriptors being used to describe patient cochlear geometry, biomimetic cochleae with similar patients' geometric descriptors can approximately capture the overall contour of the cochlear lumen which encapsulates the length of the CI array (up to 1.5 turns, $n = 3$, see corresponding analysis in Supplementary Fig. 7a). Hence, similar plain X-ray imaged electrode positions and the angular insertion depths were observed in the biomimetic cochleae and in the patients implanted with the same type of CI (Fig. 4c(ii) and Supplementary Fig. 7b). Statistically, the dependence of the CI angular insertion depth on the cochlear width was also similar, comparing the biomimetic cochlea data and the patient data (Supplementary Fig. 7c). This gives further confirmation that the 3D printed cochleae have adequate structural rigidity and anatomy to provide geometrically-guided implant insertion and positioning. It should be noted that since the 3D printed cochleae do not present the intracochlear membrane structures, the associated volume restriction effects on CI electrode positioning might not be fully captured in the 3D printed models.

Next, we acquired intracochlear EFI profiles (normalised intracochlear voltage distribution) in our cochlear models with a CI[1J] (Advanced Bionics HiRes 90 K® implant with HiFocus™ 1 J electrode) electrode array inserted. EFI samples the intracochlear voltage (V) along with the electrode array in response to a current injection or a stimulation impulse ($I_{stim}$) at each electrode (Fig. 1b). The off-stimulation measurements in EFI profiles contain information about the induced voltage spread characteristics of the cochlea. EFIs and similar measures (e.g. transimpedance matrix from

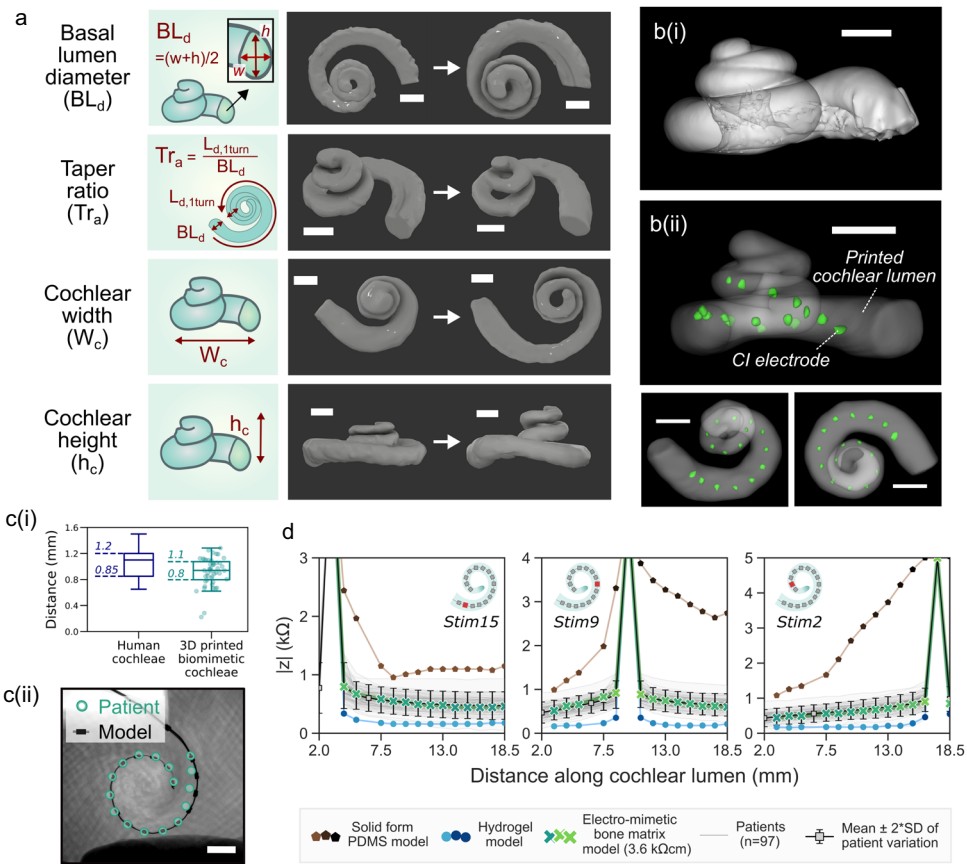

**Fig. 4 3D printed biomimetic cochleae replicate the broad anatomical spectrum of human cochleae, enable geometrically-guided CI positioning and give patient-relevant EFI profiles. a** μ-CT reconstructed images of the spiral lumen of the biomimetic cochlea with different geometric features. Scale bar = 2 mm. Four geometric descriptors are used—basal lumen diameter, taper ratio, cochlear width and cochlear height. Detailed definitions and the range of the descriptors tested in this study can be found in Supplementary Table 1. **b** μ-CT reconstructed images of (i) a cadaveric cochlea and (ii) the lumen of an exemplar 3D printed biomimetic cochlea with CI electrode array (marked green) implanted. Scale bar = 2 mm. **c**(i) The electrode-to-spiral centre distance (n = 48) of the biomimetic cochleae, compared to the electrode-to-modiolus distance of human cochleae with the same CI electrode type implanted (HiFocus[TM] 1 J electrode array), replotted from literature[31]. **c**(ii) Example showing overlapped CT and x-ray images of the CI electrode positions in a patient's cochlea and in a biomimetic cochlea that has similar geometric descriptors to the patient (n = 3, Supplementary Fig. 7b). Scale bar = 2 mm. **d** Comparison of the mean patient EFI profile (n = 97), and the EFI profiles obtained from 3D printed models made of hydrogel, solid PDMS and electro-mimetic bone matrix (3.6 kΩcm). The mean patient EFI was derived from 97 clinical EFIs that are not paired with CT information (with 91 independently acquired by Advanced Bionics® and six acquired by CI[1J] from our own repository), on the assumption that the insertion depths follow the suggested insertion depth of CI[1J]. EFIs induced by the stimulations of the basal electrode (electrode 15), the medial electrode (electrode 9) and the apical electrode (electrode 2) were shown.

Cochlear Corporation® or Impedance Field Telemetry from MED-EL®) are commonly used as part of the routine CI clinical assessment.

To further demonstrate the importance of having a realistic bone matrix resistivity in reproducing the patient EFI profile, we also fabricated models made of materials with contrasting conduction properties, hydrogels (representing the highly conductive case) and solid PDMS (representing the insulating case). Figure 4d shows the mean patient EFI profile derived from 97 patients compared with the EFI profiles of the 3D printed models with different matrix material properties. We found that the solid PDMS model led to a steeper and extremely asymmetrical EFI profile (as seen in the stimulation at the medial electrode), strongly mismatched with real patient profiles. In comparison, the conductive hydrogel model resulted in a low magnitude EFI profile, which sits outside the patient population EFI. By replicating realistic bone resistivities with electro-mimetic bone matrices, our biomimetic cochlea can be designed to match real patient stimulus spread characteristics.

**Clinically validated 3PNN shows high statistical predictive performance.** By training a neural network (NN) machine learning model with the dataset of EFI profiles acquired from the 3D printed biomimetic cochleae, a 3D printing and neural network co-modelling (3PNN) framework (Fig. 5a) was established to model the relationship between EFIs and the electroanatomical features of the CI implanted biomimetic cochleae. Reasons for using neural network modelling instead of other existing computational models are discussed in Supplementary Note 2. To support various application needs, we developed forward-3PNN and inverse-3PNN. Forward-3PNN is used when patients' cochlear geometry is known (i.e. through a preoperative CT scan), and the algorithm can predict the most probable off-stimulation EFIs arising from different electroanatomical descriptors of a cochlea. The patient-specific EFI prediction covers the initial 2–18.5 mm section of a CI electrode array from different manufacturers that may have different electrode positions and spacings. Inverse-3PNN is used when a patient EFI is given, and the algorithm can infer the most probable distribution of the

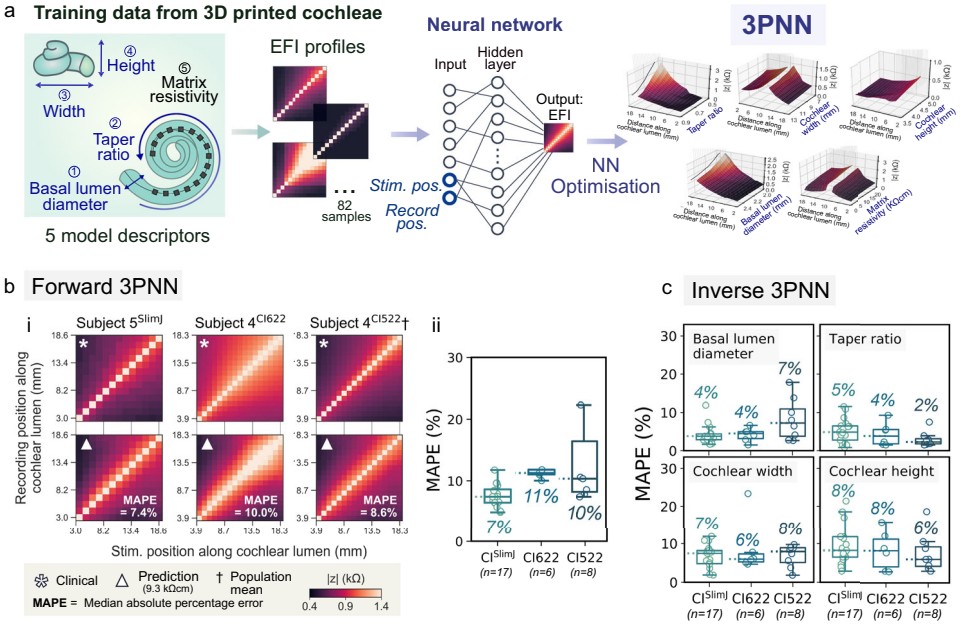

**Fig. 5 Clinical validation of 3PNN. a** Schematic of the workflow of 3PNN. 3PNN was developed by training a neural network machine learning algorithm with the EFI profiles acquired from the 3D printed biomimetic cochleae. 3PNN maps the correlation between the five model descriptors and the most probable EFI profile as a function of CI electrode position. The hyperparameters of 3PNN were tuned using tenfold cross-validation to achieve the best predictive performance (Supplementary Fig. 9). **b** Validation of forward-3PNN for predicting patient off-stimulation EFIs (matrix resistivity input = 9.3 kΩcm). (i) Representative off-stimulation EFI predictions for different CI electrode types, as compared to the corresponding clinical patient data; and (ii) boxplots summarising the overall performance of forward-3PNN, with the median MAPE of each CI electrode type indicated on the figure. Full validation results can be found in Supplementary Fig. 11. **c** Overall performance of inverse-3PNN for inferring the patients' cochlear geometric descriptors for different CI electrode types, with the median MAPE stated for each descriptor. Full validation results of inverse-3PNN can be found in Supplementary Fig. 13. In **b**(ii) and **c** the line in each box represents the median, with the box denoting the interquartile range and the whiskers denoting the ±1.5 of the interquartile range.

electroanatomical descriptors (i.e. the four geometric descriptors and the cochlear tissue resistivity) of the patient's cochlea. The broad applicability of 3PNN on different electrode types (HiFocus[TM] 1 J electrode array (CI[1J]), HiFocus[TM] SlimJ electrode array (CI[SlimJ]), Cochlear [TM] Nucleus® slim straight electrode CI622 and Cochlear [TM] Nucleus® slim straight electrode CI522) is validated in Supplementary Figs. 9–13, with its clinical predictive power demonstrated below.

We validated the clinical applicability of 3PNN using routinely acquired clinical data of different implant types. In total, 31 paired sets of patient's CT scan and EFI profile were used for validation. They were acquired using either a CI[SlimJ] ($n = 17$), a CI622 ($n = 6$) or a CI522 ($n = 8$). Here, we assumed the inputs of the stimulating and the recording electrode positions follow the manufacturers' suggested insertion depths (Supplementary Table 2) for predicting the most likely outcomes. Starting with our forward-3PNN, we predicted the patients' off-stimulation EFI profiles based on the four geometric descriptors measured from their CT scans, while taking the matrix resistivity input as 9.3 kΩcm (the mean reported resistivity of live human skulls[22–26], see Supplementary Fig. 2a). Without any model adjustment for the different CI types, 28 out of the 31 EFI reconstructions achieve a MAPE (median absolute percentage error) <12% (Fig. 5b and Supplementary Fig. 11), despite of the limited resolution of patients' cochlear CT scans, and the substitution of the unknown patient cochlear tissue resistivities with the reported mean human skull resistivity. For a selected patient (subject 4[CI522]) whose EFI profile matches the population mean EFI ($n = 97$), forward-3PNN was shown to achieve a MAPE = 8.6% for the EFI reconstruction (Fig. 5bi and Supplementary Fig. 12b, c). The capability of 3PNN

to give patient-dependent EFI predictions is confirmed in Supplementary Table 3 which cross-compares the MAPEs calculated between the patients' EFIs and the 3PNN predictions, and the MAPEs between the patients' EFIs and the population mean. Next, we validated our 3PNN by inversely inferring the distribution of the four cochlear geometric descriptors that could match a patient's off-stimulation EFI profile with a similarity >89% (Similarity (%) = 1 – MAPE (%)). Comparing the predicted distributions of the geometric descriptors with the corresponding patient's features measured from their CT scans, the median MAPE is ≤8% (Fig. 5c and Supplementary Fig. 13). The above high statistical prediction accuracy demonstrates the capacity of 3PNN to autonomously predict clinical EFIs or patients' cochlear features for different electrode types without further need to adjust the machine learning model that is trained by the dataset acquired from the CI[1J].

**Effect of cochlear electroanatomy on CI voltage spread.** With the validated 3PNN model, we proceeded to investigate how the CI voltage spread characteristics could be affected by the four geometric descriptors and the matrix resistivity. Using forward-3PNN, we simulated EFI profiles by sweeping through different combinations of the five model descriptors (examples shown in Supplementary Fig. 14). In total, we sampled 3125 ($5 \times 5 \times 5 \times 5 \times 5$) combinations to represent the entire modelling space of the five model descriptors and predicted their off-stimulation EFIs. To parameterise the voltage spread characteristics for each predicted EFI profile, we fitted a power law following Eq. (1), to each stimulus spread toward the apex and

toward the base (detailed example shown in Supplementary Fig. 15),

$$|z| = \frac{V}{I_{stim}} = A|x|^{-b} + C \quad (1)$$

$$\frac{d|z|}{dx} = -Abx^{-b-1} \quad (2)$$

where $|z|$ is the transimpedance magnitude, $V$ is the voltage between the recording electrode and the ground electrode, $I_{stim}$ is stimulation impulse current, $x$ is the distance between the stimulating and the recording intracochlear electrodes along the CI, $A$ and $b$ are fitting coefficients and $C$ is baseline constant of the EFI, which is defined as the minimum value of the EFI. Equation (1) was adopted here because, theoretically, volume conduction from a point source in a homogeneous medium should follow an inverse relationship with the form of $|z| = \frac{1}{4\pi\sigma r}$ (where $\sigma$ is the conductivity of the homogeneous medium and $r$ is the distance between the stimulating and the recording intracochlear electrode)[32], and the constant $C$ captures the baseline feature of EFIs as $|z|$ approaches the baseline when $x \to \infty$. Our goodness-of-fit test in Supplementary Fig. 16 also supports the use of Eq. (1) to describe EFI features. To quantify the slope of the stimulus spreads, we computed the derivative of Eq. (1) fitted EFI with respect to $x$ (as shown in Eq. (2)) for toward the apex or toward the base directions, and used the mean slope at the $x = 1$ mm position ($\overline{Slope}_{x=1mm}$) as an indicator of the sharpness of voltage drop toward the apex and the base of the cochlea.

As shown in Fig. 6a, we found that the voltage drop is shallower (smaller $\overline{Slope}_{x=1mm}$ value) in cochleae with larger basal lumen diameter and less tapered cochlear lumen (i.e. taper ratio closer to 1). Therefore, we predict that cochleae with these geometric features could experience broader 'current spread', which may activate neurons over a broader spatial region (thus broader spectral convolution). It should be noted, however, that the activation function for neurons should also be considered for a more sophisticated prediction of the induced firing of neurons[33]. To further evaluate the relative importance of each descriptor on EFI and its parametric fitting coefficients (i.e. $\overline{Slope}_{x=1mm}$ and the baseline constant $C$ in Eq. (1)), we performed a global sensitivity analysis (see Methods, Supplementary Fig. 17 and Supplementary Tables 4, 5). The finding suggests that the taper ratio is the most important factor affecting the sharpness of voltage drop ($\overline{Slope}_{x=1mm}$), whereas the matrix resistivity and the cochlear width are the dominant factors affecting the baseline constant ($C$ in Eq. (1)).

**On-demand creation of biomimetic cochleae inheriting patient EFIs.** The clinical validation of 3PNN demonstrates that the 3D printed biomimetic cochleae can reproduce the off-stimulation EFIs of CI users with high fidelity, despite the physical simplicity of the models. With this validated platform, we further demonstrate its application to construct on-demand cochlear models that can yield patient-specific off-stimulation EFI profiles. To do this, we first used inverse-3PNN to obtain the distribution of the model descriptors that could match each patient's off-stimulation EFI profile with an average similarity over 90% (Fig. 6b(i) and Supplementary Fig. 18). Subsequently, embedded 3D printing was used to fabricate a patient-specific biomimetic cochlea exhibiting the features of the median set of the model descriptors inferred from inverse-3PNN. As shown in Fig. 6b(ii), the EFI profiles measured from the 3D printed biomimetic cochleae show a good resemblance to their corresponding patients' off-stimulation EFI profiles, with MAPE <12%, while the patients' EFI profiles show a dissimilarity of >30% MAPE.

Beyond the application of reproducing patient-specific EFI profile with a physical 3D printed model, our platform further points to the potential occurrence of atypical EFI profiles, such as the 'mid-dip' characteristics observed in patients. The 'mid-dip' characteristic (Fig. 6c), which is distinguished by a dip in the EFI profile at the medial electrodes, has not been given a clear clinical explanation. It is uncertain whether unusual implantation orientations or patient-specific cochlear biologic properties could be the origin. By visualising the positions of electrodes in our 3D printed models with μ-CT imaging, we found that the electrode position, which was guided by the cochlear geometry, could be a potential explanation. In the model with the 'mid-dip' characteristics, the electrode positions appear to change abruptly (left panel in Fig. 6c(iii)), where electrode 8 (e8) was adjacent to two 'near-wall' electrodes (e9 and e10) that were in close proximity to the spiral centre. This sudden decrease in the electrode-to-wall distance can potentially cause a slight increase in the EFI profile, hence a dip at e8 in the profile. On the contrary, in the model without the 'mid-dip' characteristics, the electrode positions changed gradually. This suggests that the relative position of the electrode to the neighbouring electrodes and the lumen wall can be one of the causes giving rise to the mid-dip abnormality in the EFI profile.

**Informing patient-specific cochlear tissue resistivity.** As the absolute resistivity of patients' temporal bones near the cochlear vicinity cannot be measured noninvasively in living subjects, our inverse-3PNN further presents a unique capability in inferring the resistivities of patients' cochlear tissues based on their individual EFI profiles. Supplementary Fig. 19 shows the ranges of the patient-specific resistivities ($n = 37$), which were deduced with unknown geometric descriptors for subjects[1J] 1–6 and with paired preoperative CT (thus known patient geometric descriptors) for the remaining 31 subjects. All the predicted patient resistivity ranges (0.6–20.3 kΩcm) lie within the reported resistivity range of live human skulls (0.6–26.6 kΩcm)[22–26]. In particular, the mean predicted patient cochlear resistivity (6.6 kΩcm, $n = 37$) is close to the mean reported resistivity of live human skulls (9.3 kΩcm).

**Discussion**
We created a physical library of 3D printed biomimetic cochlear models that statistically captures the reported broad spectrum of off-stimulation EFI profiles of CI patients, which are dependent on the patterns of electrical conduction through tissues. The 3D printed cochlear models can be used multiple times (Supplementary Fig. 5c, d) and were designed with impedance-tuneable electro-mimetic bone matrices that display suitable mechanical stiffness for geometrically-guided CI electrode insertion while limiting damage to CI electrodes during insertion. Complementary to FEM, the 3D printed biomimetic cochleae offer a robust physical means to replicate the dynamics of ionic conduction and the electron-ion interaction in cochleae with implanted CIs. This is useful as it bypasses the sensitivity in the choice of boundary conditions that are required in FEM (Supplementary Fig. 1a), and it intrinsically captures physical phenomena that could be difficult to replicate fully in FEM.

The use of standard-of-care patient CT scans in 3PNN is practical for clinical translation because high-resolution micro-CT scans cannot be performed in living patients. As the associated resolution of clinical CT scans does not allow for detailed construction of cochlear surface contours, nor the inclusion of the membranous structures (~2 to 4 μm thick as reported in literature[10,34]), 3PNN does not aim to capture the thorough structural details of human cochleae. Several potential sources of uncertainty are noted in 3PNN. These include the discrepancy

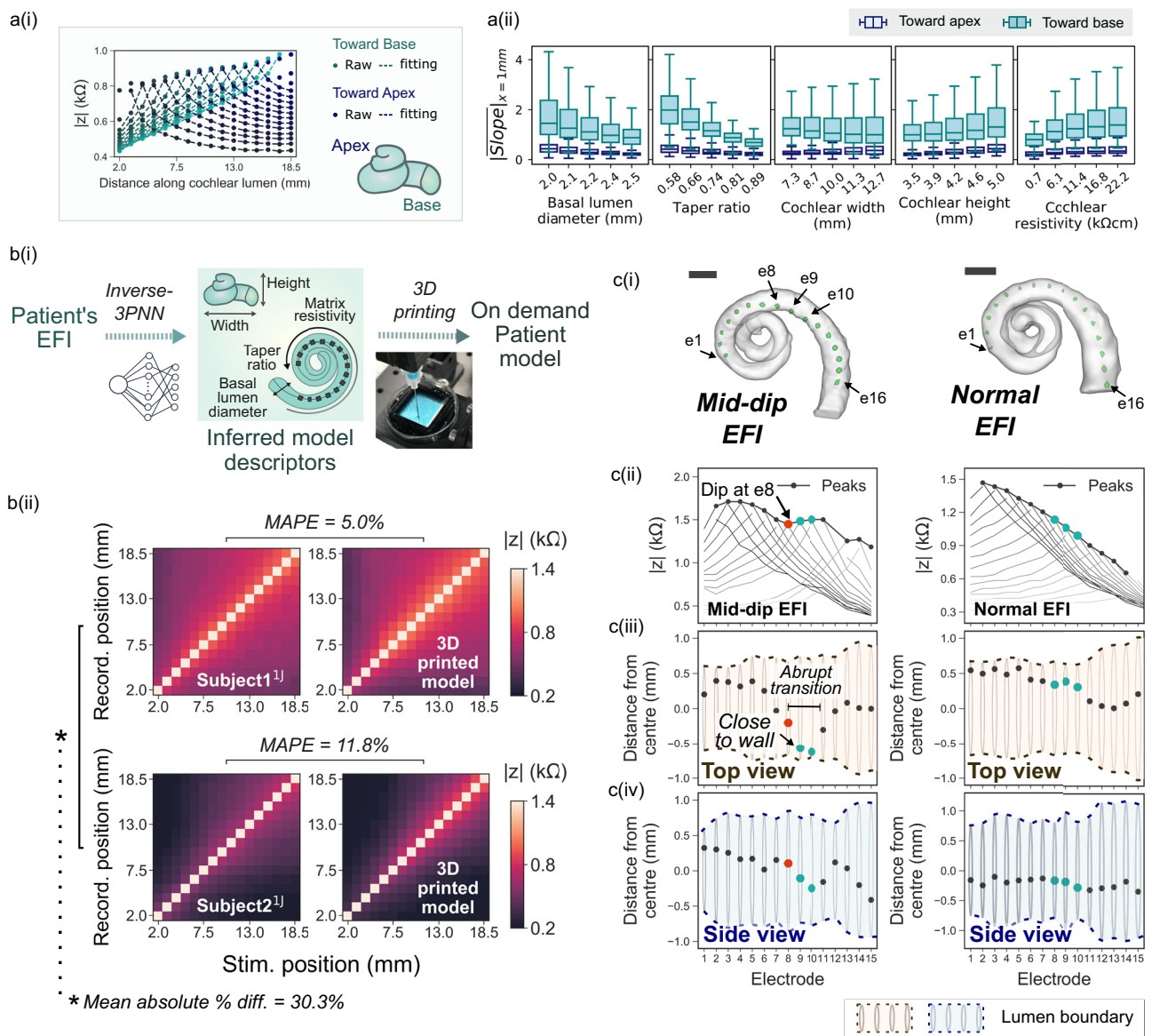

**Fig. 6 Broad applicability of 3PNN for clinical informatics. a** (i) Schematic showing the stimuli spreads towards the apex and the base of the cochleae in an EFI. (ii) The trend of $\overline{Slope}_{x=1mm}$ of the stimulus spreads toward the cochlear apex and the cochlear base across each model descriptor. The line in the box represents the median of the $\overline{Slope}_{x=1mm}$ of 625 (5 × 5 × 5 × 5) predicted samples, with the box denoting the interquartile range and the whiskers denoting the ±1.5 of the interquartile range. $n = 625$ inferred using the model descriptors sampled uniformly in the modelling space. **b** (i) Schematic showing the process to generate the patient-specific biomimetic cochlear model, where inverse-3PNN was used to deduce the distribution of the model descriptors best-fitting the patient off-stimulation EFI, and the patient cochlear model was then fabricated by 3D printing with a predicted set of the model descriptors (Supplementary Fig. 18). (ii) Comparison of the off-stimulation EFIs of two patients and the off-stimulation EFIs acquired in their corresponding biomimetic cochleae. **c** The electrode positions in a model showing an atypical 'mid-dip' EFI profile (left) and a model with a typical EFI profile (right). (i) Reconstructed 3D μ-CT volumes of the cochlear lumens of the biomimetic cochleae with a CI electrode array inserted (marked green). Scale bar = 2 mm; (ii) Off-stimulation EFI profiles of the models with the peaks indicating the maximum |z| of the spread distributions at off-stimulation positions; (iii) Top view and (iv) side view of the cochlear lumens of the models, showing the positions of the electrodes in the lumens of the models relative to the lumen wall. Distance in the negative direction refers to the distance towards the cochlear centre, vice versa. Electrode 8 (red) and electrodes 9–10 (blue) are highlighted to contrast the electrode contour which generates the mid-dip EFI.

caused by the absence of the intracochlear membranes in the 3D printed models, the uncertainties in the measurements of the low-resolution clinical CT scans, the deviations in the vertical position of CI electrode array in the cochlear lumen, the deviations in the CI electrode insertion depth due to different surgical practices and the dimensional discrepancy between the patient's cochlea and the cochlear lumen described using the four geometrical descriptors. Their potential effects on EFIs are summarised in Supplementary Table 6. In addition, the 3D printed cochleae did not account for

the frictional force generated during CI electrode insertions beneath the basilar membrane in human cochleae, which may occasionally cause electrode array buckling or even intracochlear trauma affecting CI performance[35,36]. We suggest that friction could have attributed to the localised buckling configuration of the CI electrode array captured in the 3D model giving the 'mid-dip' EFI. Future studies can explore the possibility of incorporating the membranous structures into 3D printed cochlear models, and coupling computational mechanics in the modelling process.

Adopting machine learning along with parametric descriptions of the cochlear geometry, 3PNN requires only a fraction of the computation time per EFI prediction (estimated 300 times faster) compared to our FEM models (for Intel i5 CPU). The fast and automated nature of 3PNN facilitates the generation of sufficient amount of simulated data for deciphering trend and sensitivity in a high dimensional problem. This is imperative for solving the 'volume conduction' problem, the first step in computational neuroengineering for modelling electrical stimulation in a biological structure[20]. Our work also suggests that the intracochlear excitation spread can be largely reproduced by physically replicating the volumetric conduction within the cochlear lumen and the cochlear tissue vicinity without biological components. Further studies that evaluate the correlation between the intracochlear voltage distribution and the excitation of neural cells will be of particular benefit to expand the use of 3PNN in modelling the signal perception at the neuronal level.

Our framework could potentially provide the first approach to readily infer the in vivo bulk resistivity of individual patient's cochlear bone matrix via CI telemetry. Validation of the accuracy of the cochlear tissue resistivity prediction is not performed in the current work; this is because, as of yet, there is no reported method to measure cochlear tissue resistivity in live patients. In the present work, a default resistivity value of 9.3 kΩcm (mean resistivity of a live human skull) was used to approximate the patient-specific resistivity of cochleae tissues in forward-3PNN. Thus, providing future validation to the inferred mean cochlear tissue resistivities (e.g. 6.6 kΩcm, $n = 37$) can potentially further improve the predictive power of forward-3PNN. Alternatively, future investigations which explore the correlation between the 3PNN inferred cochlear tissue resistivity and the cochlear physiological and pathological status may provide a foundation for the use of CI telemetry as a diagnostic indicator. This might enable the detection of early abnormalities after CI implantation, without resorting to imaging methodologies that use ionising radiation in patients (which particularly should be avoided in children).

Overall, 3PNN was demonstrated to be predictive for correlating the off-stimulation EFI and the geometric parameters collected from clinical patient CTs, without the need for model adjustment and re-calibration. This was validated with clinical EFI data of four different CI types (up to a position of 18.5 mm along the cochlear lumen), and 28 out of 31 predictions show good accuracy, MAPE <12% (median MAPE = 8.6%). Therefore, the co-modelling framework has the potential capability of forecasting the stimulation performance of CIs from different manufacturers, hence assisting the development of CI electrode arrays tailored to patient's cochlear anatomy. Comparing to conventional animal and cadaver models, the 'print-and-learn' modelling concept proposed here offers a physical-manipulatable, ethical and economic approach, which may help reduce the need for animal experiments. Complemented with FEM, 3PNN could form a building block for future cochlear digital twins for CI testing. With the rising usage of neuromodulating electronic implants, we anticipate that our 'print-and-learn' co-modelling concept could facilitate the physical modelling and digital twin innovation of other bioelectronic implant prototypes, beyond its applications in CIs.

## Methods

**3D printing material preparation.** The fugitive ink was prepared by dissolving 30 w/v% Pluronic F127 (P2443, Sigma-Aldrich) in a 1 w/v% NaCl (10616082, Fisher Scientific) solution. For creating 3D printed models made of PDMS or electro-mimetic bone matrices, PDMS elastomer (Sylgard™ 184 Dow, 10:1 base polymer to curing agent ratio) was used. The pre-crosslinked mixture was poured into a petri dish and degassed in a vacuum desiccator for at least 3 h prior to printing. For preparing 3D printed models made of hydrogels, hydrogels were prepared with 1 w/v% NaCl solution as the base solution according to their weight/volume concentration (w/v%) listed in Supplementary Fig. 5a. The types of hydrogels investigated were gelatin from porcine skin (G1890, Sigma-Aldrich), xanthan gum (G1253, Sigma-Aldrich), agarose (A9539, Sigma-Aldrich), gellan gum (P8169, Sigma-Aldrich); the types of fillers were talc (243604, Sigma-Aldrich), hydroxyapatite (21223, Sigma-Aldrich) and PDMS microbeads.

**Embedded 3D printing of biomimetic cochleae.** All models were fabricated with a bespoke multi-material robotic bioprinter. Five model descriptors (basal lumen diameter, taper ratio, cochlear width, cochlear height, and matrix resistivity) were used to define the model features. Prior to the fabrication process, the structure of the microchannels in the PDMS matrix was designed on Slic3R (1.3.0, slic3R.org) for tuning the void in the electro-mimetic bone matrix to achieve the desired matrix resistivity (Supplementary Fig. 3a). The correlation between the resistivity of the electro-mimetic bone matrix and its void fraction can be found in Supplementary Fig. 3e. The printing path of the microchannel structure was then converted to Gcode using Slic3R.

In the fabrication process (Supplementary Video 2), first, the sacrificial interconnected grid network designed above was embedded printed inside uncured PDMS using a 30 w/v% Pluronic F127 ink. At ambient temperature, Pluronic F127 ink at 30 w/v% retains its 3D structural integrity inside the PDMS matrix, and the interconnected network provides sufficient mechanical support for the following embedded printing of a cochlea-shaped structure. Next, a cochlea-shaped spiral was printed inside the electro-mimetic bone matrix. The printing path of this cochlea-shaped structure was defined by the four geometric descriptors and a spiral trajectory derived from the mathematical model of human cochlear geometry developed by Pietsch et al.[37]. The correlation between the dimensions of the features and the process parameters of the printer can be found in Supplementary Fig. 22. The distances between the edges of the model and the printed cochlear lumen are at least 4 mm to ensure that the boundary is far enough and will not cause any effect on the EFI measurement. The total printing time of a model ranges from 30 min to 3 h depending on the density of the embedded interconnected channels. After printing, the matrix was cured at 60 °C in an oven for 3 h and stored in a bath of 1 w/v% NaCl solution at 4 °C for dissolving the sacrificial Pluronic F127 embedded in the electro-mimetic bone matrix. The NaCl bath was changed several times to ensure that all Pluronic F127 inside the matrix was removed. In total, 82 biomimetic cochlear models with different combinations of model descriptors were fabricated. The specifications of the 82 models can be found in Supplementary Table 7.

The hydrogel and hydrogel-fillers models were similarly fabricated but without the procedure of creating the microchannel networks. The composition of the models tested in this study can be found in Supplementary Fig. 5a. The hydrogel and hydrogel-fillers solutions were heated at 40 °C during printing to maintain a liquid state. The models were then solidified at room temperature via thermal crosslinking[38].

**EFI measurements in 3D printed biomimetic cochleae.** Prior to measurement, the 3D printed biomimetic cochleae were flushed with a 1 w/v% NaCl solution to ensure no bubble was trapped in the microchannels and the cochlear lumen of the models. 1 w/v% NaCl solution was used here as it has a similar resistivity to the conductive perilymph inside human cochleae (Supplementary Fig. 3c). All EFI (or transimpedance matrix) measurements of the 3D printed models were obtained using either an Advanced Bionics (AB) HiRes 90 K® implant with HiFocus™ 1 J electrode array (CI[1J]), an Advanced Bionics HiRes™ Ultra implant with HiFocus™ SlimJ electrode array (CI[SlimJ]) or a Cochlear™ Nucleus® Profile with a slim straight electrode (CI522). Both CI[1J] and CI[SlimJ] have 16 electrodes in total with electrode 1 being the apical-most electrode and electrode 16 being the basal-most electrode. CI522 has 22 electrodes in total with electrode 22 being the apical-most electrode and electrode 1 being the basal-most electrode. The electrode array was inserted in the cochlear lumen of the model until the distal marker of the electrode array was positioned at the lumen opening of the model, as illustrated in Supplementary Fig. 23, and the model was placed on top of the extracochlear case ground of the CI (known as the 'case ground' of CI[1J] and CI[SlimJ], or the 'MP2 plate extracochlear electrode' of CI522). The EFI profiles were acquired using the telemetry function of the CI with either the AB Volta version 1.1.1 software (research only) or Custom Sound® EP 5.1 (with research option) using the default stimulation and recording settings. The default stimulation and recording setting used in AB Volta software is a biphasic pulse with pulse width and amplitude of 36 µs (equivalent Fourier fundamental frequency ~14 kHz) and 32 µA, and a maximum sampling rate of 56 kHz, whereas Custom Sound® EP 5.1 employs a setting of a biphasic pulse with pulse width and amplitude of 25 µs (equivalent Fourier fundamental frequency ~20 kHz) and 125 µA respectively. During the acquisition of EFI, each electrode was activated individually at a time in monopolar mode, and subsequently, other electrodes measured the resulting voltage at their positions. All electrodes on the electrode array were activated one-by-one to generate the entire EFI profile. Electrodes 12 and 16 of the CI[1J] electrode array were missing as received, but this does not affect the measurements of other electrodes and the general shape of the EFI profile. For all the data presentations, the on-stimulation EFI data (contact impedance) were not compared, due to the fact that on-stimulation EFI data is dominated by the electrode interface resistance[9,39] and do

not inherently reflect the electroanatomical characteristics of human cochleae (or the 3D printed biomimetic cochleae); and on-stimulation EFI data varies over time[40] and among different CIs.

### Resistivity measurements

*Resistivities of NaCl solutions, hydrogels and hydrogel-fillers matrices.* Impedance properties of NaCl solutions at various concentrations (Supplementary Fig. 3c), hydrogel and hydrogel-fillers matrices (Supplementary Fig. 5a) were measured using a four-terminal configuration with Solartron 1260 impedance analyser and SMaRT 3.0.1 software. In this configuration, the current was passed through the sample using two $1.25\ cm^2$ square electrode plates, and the voltage was measured using two separate inner electrodes. Resistivity was converted from the plateau impedance magnitude using the following relation,

$$\rho = |z|\frac{A}{d} \tag{3}$$

where $\rho$ = the resistivity of the sample (plateau value), $|z|$ = the plateau impedance magnitude, $A$ = the area of the electrode plate in contact with the sample, and $d$ = the spacing between the two inner electrodes, which was 8.4 mm here.

*Resistivities of electro-mimetic bone matrices.* The resistivity of the electro-mimetic bone matrix $\rho_{matrix}$ associated with the plateau impedance magnitude (~300 Hz–100 kHz) was determined using the transmission line method ($n \geq 3$). In this method, each sample was segmented into at least four segments. The impedance of each segment was obtained using a two-terminal configuration with a Solartron analyser, and the width of each segment was measured. The total impedance $Z_{tot}$ can be expressed by $Z_{tot} = Z_c + Z_{sample}$, where $Z_c$ is the contact impedance between the electrode plates and the samples, and $Z_{sample}$ is the impedance of the sample. The plateau value of the total impedance magnitudes $|Z_{tot}|$ of the segments were therefore plotted against the widths of the segments $L$, and a linear regression was then used to fit the experimental data (see Supplementary Fig. 3d). $\rho_{matrix}$ was determined by multiplying the gradient of the linear regression $\frac{\partial|Z_{tot}|}{\partial L}$ with the area of the electrode plate in contact with the sample $A$, denoted as follows:

$$\rho_{matrix} = A\frac{\partial|Z_{tot}|}{\partial L} \tag{4}$$

### Electrochemical impedance spectroscopy (EIS) measurements. Electrochemical impedance spectroscopy measurements of a human cadaveric cochlea in a head, and 3D printed cochlear models made of hydrogel and electro-mimetic bone matrix were carried out using an impedance analyser (RS PRO LCR-6100) with a three-terminal configuration[41]. The measurements were taken at frequencies ranging from 10 Hz to 100 kHz, which covers the most common operating frequencies of CIs.

### Micro-computed tomography scans of the 3D printed biomimetic cochleae. CT scans of samples were acquired using a micro-CT microscope (ZEISS Xradia 510 Versa) with the following scanning parameters: Source filter LE2, tube voltage 80 kV, tube current 88 mA, exposure time 2 s, Bin 2, image taken 1024 and pixel size 17.8 μm. The volume rendering of the samples was carried out using 3D Slicer (Version 4.10.2, www.slicer.org/[42]). The dimensions of the samples were measured using the measurement tool in 3D Slicer.

To evaluate the positions of electrodes in the samples and to avoid the image distortion caused by the metallic artefacts from electrodes, pre- and post-insertion CT scans of the samples were acquired. CT volume of the cochlear lumen of the sample was rendered from the pre-insertion CT scan where there is no metallic artefact, whereas CT volume of the electrode array in the sample was rendered from the post-insertion CT scan. The two CT volumes were then aligned, and the relative position of each electrode from the centre of the cross-sectional plane of the cochlear lumen was measured using ImageJ. The 2D images of the electrode array inside the cochlear lumen of the samples were acquired using the following parameters: tube voltage 80 kV, tube current 88 mA, exposure time 5 s, Bin 2 and pixel size 25.6 μm.

### Patient EFI profiles and CT scans. The use of anonymous patient EFI profiles with or without paired CT scans in our study was approved by the University of Cambridge Human Biology Research Ethics Committee (HBREC.2019.42) and the Cambridge Biomedical Research Centre (Ref: A095451). Informed consent from the human participants is not required for this study as the clinical data used here are retrospective and anonymous. In total, 128 clinical intra-operative EFIs (also known as transimpedance matrix profiles) were used in this study. Of the 128 profiles, 91 profiles (without paired CT scan data) were kindly provided by Advanced Bionics® and the rest were obtained from 37 anonymous patients (31 with paired CT scan data and 6 without paired CT data) who have undergone cochlear implantation at the Emmeline Centre for Hearing Implants in Cambridge, UK. As the implant types of the EFIs provided by Advanced Bionics® are not known, their insertion depths were assumed to be equal to the suggested insertion depth of the HiFocus™ 1 J electrode array. The 37 anonymous EFI profiles acquired in our centre were randomly chosen to represent the variation in the

patient data without CT scans ($n = 97$) (Supplementary Fig. 12a). Out of the 37 EFI profile data sourced from our centre, six profiles were acquired from the Advanced Bionics HiRes 90 K® implant with HiFocus™ 1 J electrode array, 17 profiles from the Advanced Bionics HiRes™ Ultra implant with HiFocus™ SlimJ electrode array, six profiles from the Cochlear™ Nucleus® Profile Plus with slim straight electrode CI622 and eight profiles from the Cochlear™ Nucleus® Profile with slim straight electrode CI522. These EFI profiles were collected using the telemetry function of the CI with either the AB's Volta 1.1.1 software (research only) and the Custom Sound® EP 5.1 software (with TIM research option) using the default stimulation and recording settings.

Thirty-one CT scans of the patients (which had paired EFI) implanted via the round window approach with either a HiFocus™ SlimJ electrode array ($n = 17$), a Cochlear™ Nucleus® CI622 electrode ($n = 6$), or a Cochlear™ Nucleus® CI522 electrode ($n = 8$) were used in the validation of 3PNN. They were obtained as part of the routine preoperative assessment at our centre, and were acquired in helical scan mode using Siemens scanners (Siemens Flash and Siemens Definition AS) with a tube voltage of 120 kV and automatic tube current ranging from 139 to 214 mA. The images were reconstructed at a resolution of 0.4 mm × 0.4 mm × 0.4 mm using Siemens 80 u bone reconstruction algorithm in an axial plane.

### Development of 3PNN. 3PNN was developed by employing a multilayer perceptron (MLP), a class of feedforward artificial neural network (NN), to learn the mapping from the inputs (the five model descriptors of the biomimetic cochleae, the stimulus position and the recording position) to the outputs (EFI, also known as transimpedance matrix profiles) (see Fig. 5a, for detail of the choice of the model see Supplementary Note 2). An MLP model is a fully connected network that consists of an input layer, hidden layers and an output layer of perceptrons (or nodes), and by varying the weight of how each of the nodes are connected, it approximates the complex relationship between the inputs and the output[43]. The activation function of the nodes was chosen to be the rectified linear unit (ReLU) function. Tensorflow[44] (version 2.1.0), an open-source Python library, was used to construct the MLP models. 3PNN was trained using backpropagation with the Adam stochastic optimisation method[45]. Since 3PNN was developed based on the EFI profiles acquired by AB HiFocus™ 1 J electrode array with electrodes at 2–18.5 mm along the cochlear lumen[46], the predictable positions of EFIs are 2–~18.5 mm along the cochlear lumen.

The performance of NN models depends on a good setting for hyperparameters, a grid search varying the number of hidden layers from 1 to 10 (1, 2, 3, 5, 10) and nodes from 16 to 64 (16, 24, 32, 64) was performed to determine the best performing hyperparameters (see Supplementary Fig. 9 for detail of the hyperparameter tuning). The best performing hyperparameters were defined as the hyperparameters that yield the highest average $R^2$ score and the smallest average median absolute percentage error (MAPE) in tenfold cross-validation[47]. We found that the model trained with 1 hidden layer and 32 nodes has the highest average $R^2$ score (0.87) and the smallest MAPE (11.9%). After tuning the hyperparameters, the 3PNN was retrained on the full dataset with the best performing hyperparameters to produce the final model.

The inverse prediction was carried out by the Approximate Bayesian Computation-Sequential Monte Carlo (ABC-SMC) algorithm[48]. ABC is a computational framework under Bayesian statistics that uses a sequence of intermediate threshold [$\varepsilon_0 > \varepsilon_1 > \varepsilon_2 > \varepsilon_3 > \ldots. > \varepsilon_f$] to converge towards the optimal approximate posterior distribution through a number of intermediate posterior distributions. Here, the algorithm infers the distribution of the model descriptors that leads to an EFI profile with a MAPE less than a predefined threshold ($\varepsilon_f$) to the given EFI profile. $\varepsilon_f$ was determined as the smallest MAPE the programme could reach from the previous threshold level within 2 h when running the programme with a threshold sequence from 20 to 2% in increments of 0.5% (predictions with unknown geometric descriptors) or 0.1% (predictions with known geometric descriptors), which is subject to the noise level of the data. To approximate the final posterior distribution (which does not have a closed-form expression), for each inverse prediction, 1000 samples of the posterior distribution of the model descriptors were plotted. PINTS[49], an open-source Python package, was used to perform the inference and sampling.

### Clinical predictions of 3PNN. As this study aims to predict the most likely EFI outcomes, in all predictions, the stimulating and the recording electrode positions were assumed to follow the CI specification, as shown in Supplementary Table 2. In the validation of forward-3PNN, patients' model descriptors measured from their CT scans and the mean reported resistivity of live human skulls (9.3 kΩcm) were used as the inputs in the forward predictions of patient EFIs. EFI arising from off-stimulation positions up to 18.6 mm along the cochlear lumen were predicted and compared with the corresponding EFI measurements acquired in patients. Each forward prediction takes ~0.4 s. For all inverse predictions performed in this study, patients' model descriptors were predicted using their off-stimulation EFI profiles up to 18.6 mm along the cochlear lumen. Supplementary Table 8 summarises the values of the final MAPE threshold, $\varepsilon_f$, used in the inverse predictions in this study.

### Production of 3D printed models that give patient-specific EFI profiles. Two extreme on-demand 3D printed models that give patient-specific EFI profiles were

fabricated using the medians of the predicted model descriptors acquired from inverse-3PNN, as stated in Supplementary Fig. 18 (matrix resistivity 6.5 versus 0.7 kΩcm, taper ratio 0.95 versus 0.71, basal lumen diameter 2.4 versus 2.3 mm, cochlear width 9.6 versus 11.8 mm and cochlear height 4.3 versus 3.9 mm). The EFIs of the models were measured using a HiFocus™ 1 J electrode array, which is the same type of electrode implanted in the patients.

**Sensitivity analysis for 3PNN.** Sobol's method[50], a global sensitivity analysis technique for nonlinear models, was employed to investigate the contribution of each 3PNN model descriptor to the model output (EFI) and its summary statistics (Supplementary Fig. 17 and Supplementary Tables 4, 5). A total of $1.68 \times 10^5$ samples of model descriptors were generated using Saltelli's sequence[51]. On top of the EFI output, for the ease of interpretation, two summary statistics were analysed in this study; they are the baseline (the coefficient $C$ in Eq. (1)) and the slope at $x = 1$ mm (the coefficient product $Ab$ in Eq. (2)) of the EFI outputs. The sensitivity of each model descriptors on the EFI and its summary statistics were quantified using the Sobol first-order sensitivity indices ($S_i$), which describe the contribution to the variance of the EFI or its summary statistics caused by one model descriptor only; the second-order indices ($S_{ij}$), describing the contribution to the output variance due to the interaction of two model descriptors; and the total-order indices ($S_T$), measuring the all order effect contribution to the output variance for each model descriptor. The analyses were performed using a Python open-source package SALib[52]. Full results of the Sobol sensitivity analysis are available from the GitHub repository[53].

**Statistical method.** Median absolute percentage error (MAPE) was chosen as the error measure in this study because it presents the percentage change due to the error and avoids being too sensitive to outliers. The MAPE between the predicted EFIs (EFI$_{\text{pred}}$) and the experimental EFIs (EFI$_{\text{exp}}$), and the MAPE between the predicted geometric descriptors and the actual CT-measurements were evaluated using Eqs. (5) and (6) respectively, where $a_{ij,\text{exp}}$ and $a_{ij,\text{pred}}$ are the entries in EFI$_{\text{exp}}$ and EFI$_{\text{pred}}$, and $G_{\text{CT}}$ and $\{G_1, G_2, \ldots, G_{1000}\}$ are the CT-measured geometric features and the 1000 predicted geometric features. Similarity is defined by Eq. (7).

$$\text{EFI}_{\text{exp}} = \begin{bmatrix} a_{11,\text{exp}} & \cdots & a_{1j,\text{exp}} \\ \vdots & \ddots & \vdots \\ a_{i1,\text{exp}} & \cdots & a_{ij,\text{exp}} \end{bmatrix} \quad (5.1)$$

$$\text{EFI}_{\text{pred}} = \begin{bmatrix} a_{11,\text{pred}} & \cdots & a_{1j,\text{pred}} \\ \vdots & \ddots & \vdots \\ a_{i1,\text{pred}} & \cdots & a_{ij,\text{pred}} \end{bmatrix} \quad (5.2)$$

$$\text{MAPE}_{\text{EFI}} = \text{median of} \left\{ \frac{|a_{11,\text{pred}} - a_{11,\text{exp}}|}{a_{11,\text{pred}}}, \frac{|a_{12,\text{pred}} - a_{12,\text{exp}}|}{a_{12,\text{pred}}}, \cdots, \frac{|a_{ij,\text{pred}} - a_{ij,}|}{a_{ij,\text{pred}}} \right\} \times 100\% \quad (5.3)$$

$$\text{MAPE}_{\text{geometricfeatures}} = \text{median of} \left\{ \frac{G_1 - G_{\text{CT}}}{G_{\text{CT}}}, \frac{G_2 - G_{\text{CT}}}{G_{\text{CT}}}, \cdots, \frac{G_{1000} - G_{\text{CT}}}{G_{\text{CT}}} \right\} \times 100\% \quad (6)$$

$$\text{Similarity(\%)} = 100(\%) - \text{MAPE(\%)} \quad (7)$$

**Reporting summary.** Further information on research design is available in the Nature Research Reporting Summary linked to this article.

## Data availability
The raw data of the 3PNN validation, Sobol sensitivity analysis, the stimulus spread trend, the resistivity prediction and the uncertainty sensitivity analyses have been deposited in Github and in Zenodo under accession code 5353394[53]. Other data generated in this study are provided in the Source Data file. Restrictions apply to the availability of the clinical EFI and CT scan data, as non-restricted access to this information could compromise patient confidentiality. These data will be made available upon request to the corresponding authors and in compliance with the ethical guideline used in the current study. Source data are provided with this paper.

## Code availability
The code of the neural network model and the Sobol sensitivity analysis used in this study are available on Github and through Zenodo[53].

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

## Acknowledgements

This work was supported by the European Research Council (ERC-StG, 758865), the Cambridge Hearing Trust and the Evelyn Trust. I.M.L. acknowledges the financial support from the W.D. Armstrong Trust and the Macao Postgraduate Scholarship Fund. C.J. acknowledges the support from the Wellcome Trust (204845/Z/16/Z). C.L.L. acknowledges the support from the University of Macau via a UM Macao Fellowship and the Clarendon Scholarship Fund. S.R.d.R. acknowledges the financial support from the Baroness de Turckheim Fund, Trinity College Cambridge. The authors acknowledge the Henry Royce Institute Cambridge Equipment (EP/P024947/1). We thank Advanced Bionics Corporation® for providing cochlear implants, software and anonymous EFI profiles on this research, Patrick Boyle for providing information on the EFI measurement setting, Anthony Dennis for technical assistance with μ-CT imaging, Tomasz Matyz for his help with the anonymised patients' CT scans, and HaoTian Harvey Shi and Ruishan Liu for test running the code.

## Author contributions

I.M.L., C.J., M.B. and Y.Y.S.H. conceived the project. I.M.L. performed the majority of the experiments and data analysis including 3D printing, creation of biomimetic cochlear EFI library and COMSOL simulation. C.J. performed the EIS experiments. C.L.L. developed the neural network model, performed the Sobol sensitivity analysis and assisted with part of the data analysis. I.M.L. performed the neural network model optimisation and simulations. C.J., S.R.d.R., Y.C.T., C.S. and M.B. assisted with the anonymised patient's data collection. Y.Y.S.H., G.G.M. and M.B. provided result interpretation. M.P.F.S. performed the mechanical deformation analysis. I.M.L. and Y.Y.S.H. wrote the initial manuscript. Y.Y.S.H and M.B. supervised the project. All authors discussed the results and revised the manuscript.

## Competing interests

M.B. received research funding from Advanced Bionics®, Cochlear Corporation® and in-kind contributions from MED-EL® on other research areas but not on the present study. The remaining authors declare no competing interests.
