## [Peer Review File · Nature Communications]

Reviewers' Comments:

Reviewer #1:

Remarks to the Author:

The paper "A neural network trained by 3D printed biomimetic cochleae provides clinical informatics for cochlear implant patients" presents the development of a neural network model that is informed by measurements in 3D printed biomimetic cochleae with realistic conductivity and anatomy. The model uses 4 geometric features to predict the intracochlear voltage (electric field imaging; EFI). The model has been also used to infer the cochlear geometries based on the measured EFIs.

The research presented in the manuscript is highly innovative and covers a wide range of challenging methods including neural networks, electric field measures in vivo, 3D printed models, finite element method, image analysis through CT and CBCT and a few more. The methods used are highly complex and require a high level of expertise. Unfortunately, the validation of the model predictions in human CI users cannot keep up with this quality. For instance, the 3PNN model has only been validated in 6 subjects with 1 electrode type. Moreover, these 6 subjects had very similar EFI profiles as well as similar geometry descriptors (Fig.5bc).

The option to "infer" the in vivo bone resistivity (which is advertised in the abstract and forms a substantial part of the discussion) is not validated at all.

The reviewer has some general concerns and a list of detailed corrections presented below.

General Comments

- In general, the methodology presented in the manuscript is innovative and promising and well presented with an exceptional level of detail. The quality of both engineering and scientific work in developing the methodology seems to be very high. Even if it may be logical to think that the spread or width of the EFIs may be related to speech understanding performance, there is no much evidence in the literature for this (see for example: Jürgens et al 2018, PlöseOne). In my opinion the third line in the abstract should be weakened or look for other alternative main motivations for modeling EFIs.
- It has been previously reported that EFIs or transimpedance measurements relate to electrode angular depth of electrode insertion (Aebischer et al. 2020, IEEE). Insertion depth depends on different factors including cochlear anatomy (length and shape), surgical procedure or even surgeon. Moreover, it is possible that impedances are affected by electrode-modiolar distance (e.g. Saunders et al 2002, Ear Hear). In your neural network and in your models you did not study the effect of different electrode locations for the same cochlea on the EFI. In other words, two subjects having the same exact cochlea anatomy may receive the cochlear implant electrode contacts in different locations which would lead to different EFIs, however based on your model the same EFI would be predicted for both subjects as only cochlear anatomical features are used.
- Include supplementary table with specs of the 82 printed biomimetic cochlea models
- Why do you use different threshold criterions for the backward 3PNN model? E.g. 8% MAPE (line 652), 6% MAPE (line 660), 10% MAPE (line 668) etc.
- Not clear how the impedance recordings were conducted. According to Vanpoucke et al (2004), IEEE, the measurement electronics of the AB device may take samples at a rate of 56 kHz. How were you able to sample at 83 kHz? In clinical impedance measures only one value during each stimulation pulse is recorded to characterize the so-called "impedance", so that the frequency of stimulation/recording does not play a role. However, in your impedance spectroscopy you used an 83 kHz frequency, did you use here analog stimulation? How does your impedance spectroscopy relate to the EFI measures with the CI?
- Effect of reference electrodes in the impedance measures. Typically, EFIs are measured using a stimulating electrode and a reference electrode for the stimulation and a recording electrode and a reference electrode for the recording. Which electrodes did you use in your methods and how much were the measurements affected in your model by the fact that you do not have a head in your cochlear printed models?
- The limited characteristics or input features used in the 3PNN to predict EFIs cause that the applications mentioned in the discussion sound too optimistic at this stage of the research. Modeling of tissue growth, ossification of the cochlea and different electrode locations should be incorporated into the model to be able to successfully use the 3PNN for the mentioned clinical applications.

- Sometimes you give a value followed by the units and sometimes followed by space and the units. Check format.

Detailed Comments

Abstract

Line 20-23 "(...) the model can reconstruct patients' clinical electric field imaging (EFI) profiles arising from off-stimulation positions with a 90% mean accuracy (n=6)."

Only shown for slimJ users with very similar EFI profiles. It would be interesting to see the accuracy for different electrode types and more varying EFI profiles.

Line 25-27 "This work (...) directly reveals individual patient's in vivo cochlear tissue resistivity (0.6 – 15.9kΩcm, n=16) by CI telemetry."

◇ No, this method is not validated. At most, you can claim that your model is able to "estimate" the resistivity, but it is not "revealed".

Introduction

Line 32 "restore electrical functions" typically a CI is used to restore "neural function" rather than "electric function". Even if neural function is based on electric function, electric seems to be too simple.

Line 35 Reference 4 is about optogenetics... sounds inappropriate here.

Line 38 Unfortunately, the CI does not preserve the tonotopy completely. There are large misalignments between the electrode locations and the frequencies associated to the electrodes. Maybe rewrite slightly this sentence. Maybe something like CI try to mimic the tonotopical structure of the cochlea ...

Line 63 Cochlear implants are typically classified as neural implants rather than electric implants.

Line 67 "auditory nerve fibers" instead of "auditory nerves".

Results

Line 132 "...the operating frequencies of CIs (i.e. f=1-100kHz)"

◇ should this be 1 Hz – 100 kHz or 1 kHz – 100 kHz?

Line 148 Here you measure the resistivity matrices at 83 kHz. I do not understand how this 83 kHz frequency relates to the EFI recorded through the cochlear implant. In the cochlear implant, the EFI is typically recorded with single pulses and only one sample for each stimulating pulse is recorded. This requires clarification.

Fig. 4a, b In the subfigures below these subpanels, I was wondering where is the round window placed? And whether different electrode locations (electrode insertion depth and electrode insertion trajectories) are possible for the same cochlea. These different electrode locations would result in different EFIs.

Fig.4d Don't need the "doubled" panels

Fig. 4d Not sure if in these plots the impedance of the stimulating electrode is also plotted.

Actually it seems that it is plotted given the large impedance at the location corresponding with the stimulating electrode. If so, why is the electrode impedance of the stimulating electrode shown here if this is not used in the models?

Line 281-283 How do you define the "similarity"?

Fig.5b The results do not look very patient-specific, maybe because of the measurements that are also very similar.

However, it would be nice to compare the performance of the 3PNN predictions against an "average" EFI profile and/or to compare predicted EFIs with EFIs from other subjects.

Fig.5c i) Color-coding is not visible

ii) Are the huge error bars for CT-measured features correct?

Line 304 Predictions are based on data from 6 subjects, where is the 1000 predicted set obtained from?

Line 315 Here you present Equation 1 which defines a function to fit the EFIs. It is however well known that the EFIs do not go to zero and have a sometimes a constant ground or base level which needs to be taken into account to fit properly the potential function shown in Equation (1). Did you observe a constant base level in your EFIs? How did you account for it in the fittings of A and b?

Line 322 I would define "r" as "the distance between the stimulating and the recording intracochlear electrode" to be more specific

Lines 322-225 Also in Fig 6. In the EFIs of the apical and basal electrodes it is not possible to fit any potential function towards the apex or toward the base respectively. Did you remove these fits

for the analysis of the variables A apex A base, b apex and b base?

Line 337 This may be a typo: neurones \diamond neurons

Fig.6a What do the marker colors stand for (color bar missing)?

Fig.6d I do not understand the "electrode-distance-distance" plots in (i) as well as the plots in (iii)/(iv)

Fig.6d ii and iii Not clear what are the peaks. Are these the peaks of the second diagonal? Peaks need to be defined

Fig.7 Color-coding for predicted resistivity is not needed as it is shown at the y-axis.

Line 394-422 The "deduction" (I would rather call it "fitting") of patient-specific bone resistivities lacks validation. This is rather a hypothesis and not a method, this should be pointed out more clearly.

Discussion

Line 431-432 What about the surgical procedure to insert the cochlear implant, or even the fact that different surgeons insert the same implant differently? An extreme case would be cochleostomy vs round window approach which theoretically should have significant effects on the electrode locations and EFIs which your model do not take into account. It would have been much better the inclusion of different electrode arrays from different manufacturers and having different insertions in the same cochlea to have more variability in measured EFIs.

Line 438 Why the electrode locations were not required?

Line 441-442 Two different electrode arrays having different dimensions, including inter-electrode spacings (1.3 mm vs 1.1 mm as acknowledged in line 648), were used for training and testing the algorithm which adds more difficulty to the aim of predicting the EFIs.

Line 462-465 Your model does not consider ossification of the cochlea, tissue growth across the electrode array and effect of electrode location. In my opinion, the potential application of the 3PNN for the applications here mentioned seems very optimistic at this stage.

Methods

Lines 531-534 Are you sure that the acquisition was done at 83 kHz? I though the AB can record at most at \sim 50 kHz? Which type of stimulation was used to measure the EFIs, analog stimulation or biphasic pulses? Most manufacturers only use biphasic pulses and clinically only one sample for each stimulation pulse is available. Which reference electrodes were used for stimulation and recording and how do these reference electrodes affect the EFI? How does this relate to the position of the reference electrode in the printed cochlear models?

Line 551 Which spacing d has been used?

Line 552 Are you sure that the stimulation rate in one electrode was 83 kHz? Again typically, biphasic pulses are used to record impedances and definitely at much lower rates. Authors should explain in detail how this fast stimulation or recording rate (not clear) was generated with the AB implant.

Line 596-598 "The EFI profiles were collected using the telemetry function of the CI with the AB's Volta software as part of the intraoperative evaluation at our centre."

\diamond what about postoperative impedance changes ?

Line 603 I am a bit concerned about the quality of the CT scans and the variability of the derived anatomical measures. How much did the resolution of the CT scan (0.4x0.4x0.4 mm³) impact the anatomical measures of the cochlea? At least you will have an error of half voxel size. Specially looking at Fig.5 in supplementary material Section II) the resolution of the human CBCT scans looks quite limited.

Line 621-621 "(..) a grid search varying the number of hidden layers from 1 to 10 (1, 2, 3, 5, 10) and nodes from 16 to 64 (16, 24, 32, 64) was performed to determine the best performing hyperparameters."

\diamond are these values appropriate? Specially when increasing the number of layers to 3,5,10 it seems that the number of parameters is extremely large in comparison to the small dataset used for training.

Supplementary information

Supp. line 94-97 "The rationale for using a 1w/v% NaCl solution is that 1w/v% NaCl solution at ambient temperature has the most similar conductivity to human cerebrospinal fluid, which is 1.79 S/m at body temperature¹⁴, and it is known that the electrical conductivity of cerebrospinal fluid is similar to the conductivity of perilymph."

\diamond provide reference

Supp. Fig.3b Why such a large difference between Y2(C) and Y2(E) ?

Supp. Fig.4a The lowest resistivity reported for live human skulls should be 0.6 k Ω cm, not 500 Ω cm – see Supp. Fig.1b and main text (lines 114, 135).

Supp Fig.5 On the small subplots you show the real CT scans. The upper plots show the cochlea after doing image processing (maybe segmentation and some filter for enhancing). How much did these filtering affect the estimation of anatomical dimensions? when enhancing edges and contours of the CBCT scans you modify the grey scale that gives information about the bone. These questions are related to the concerns about the resolution of the scans to estimate the detailed anatomical features of the cochlea.

Supp. Note 1 i) Line 174: explain ν (Poisson's ratio)

ii) For the "Global deformation" part, it is difficult to follow the calculus because the equations are unnecessarily rearranged multiple times.

For instance, use $\gamma=F/AG$ instead of $G=F/A\gamma$. The equation in line 175 is not needed then.

iii) Line 182-184: "Therefore, as the global shear strain and local normal strain caused by CI electrode array insertions are insignificant, we expect that the insertion of CI electrode (...) will not impose any significant deformation to the matrix."

◇ what is your criterion to call these values insignificant? How large would the strain have to be in order to be significant?

Supp. Fig.6ab It is very difficult to see the electrode array in these plots

Supp. Fig.6e Which boundary condition is applied here? (compare Supp. Fig.7)

Supp. Fig.7 The influence of the boundary condition is very interesting. What are the consequences? Which boundary condition fits best to the in vivo or the in vitro (3D-printed cochleae) cases? This needs to be addressed in the discussion.

Supp. Fig.8 Figure title: "Electrode-to-spiral centre distance"

Supp. Fig. 11c Provide the dimension of the input and output layers.

Supp. Tab.1 3rd row, 2nd column: should be "Ld,1turn"

Supp. Fig.9 This figure is not referenced at all

Supp. Fig.13cd i) indicate units (mm) on x-axis in panels c,d (can be confused with electrode number in a,b,e)

ii) it seems that only spread distributions with a minimum of 4 data points have been fitted

Supp. Fig.14 Flip y-axes?

Supp. Fig.15 Line 395-396: How did you select the values for the 3D printed model? Some of them are very off-centered. The simplest way would be to use the medians.

Reviewer #2:

Remarks to the Author:

The authors are to be commended for their timely and innovative work. Cochlear models would be very useful for improving cochlear implant electrode design, surgical planning, and basic research to name a few possibilities. To my knowledge, the work is novel and has potential to impact hearing science and patients in the future. Many of my questions focus on the 3D printed models that guide the development of the neural network.

1. Line 92-93: I wasn't clear if your modeling approach resolves the issues concerning boundary conditions. If so, can you point me to where the boundary conditions are discussed for your approach.

2. Lines 165-170: Great care was exercised in designing the matrix to ensure a CI electrode can be inserted without damage to the electrode. If models are to be used in a patient-specific framework, are the final positions similar to those in patients?

My concern is that later (lines 188-190) we are told that intracochlear structures (e.g., basilar membrane) are not modeled. Does the presence of intracochlear structures not affect electrode insertion? If it does not, please provide evidence.

3. The modeling done in this work is quite complex and involves a number of approximations and assumptions. It would help me if sources of errors in modeling could be tabulated in one place and their effects could be quantified. As an example, cochlear geometry is modeled using a few parameters. How close are these models to that of the patient being modeled? Global and local

metrics as used in the following article would be a nice way to quantify these discrepancies:

Noble JH, Labadie RF, Majdani O, Dawant BM. Automatic segmentation of intracochlear anatomy in conventional CT. *IEEE Trans Biomed Eng.* 2011;58(9):2625-2632.
doi:10.1109/TBME.2011.2160262

An error map as in Figure 8 of that paper would be useful.

If there are discrepancies between the model geometry and the patient's geometry as revealed by imaging, how do these propagate to errors in EFI simulations?

Similarly, other assumptions and approximations should be considered. A sensitivity analysis would help tremendously.

4. It is mentioned that the model can reconstruct patients' EFI profiles arising from off-stimulation positions with a 90% mean accuracy ($n = 6$). From a clinical perspective, what is a sufficient level of accuracy and what is the justification for this level? The accuracy of EFI should be discussed to help the reader appreciate why it is a suitable technique for testing of the model.

Also, does $n = 6$ provide sufficient statistical power in light of the large variability in cochlear anatomy?

5. Line 285: It is noted here that "the predictive quality in cochlear height is less satisfactory compared to ...". How could prediction of cochlear height be improved?

6. Line 592: It is noted that EFI profiles were chosen randomly from 16 adults. Do these 16 capture the range of variability seen in EFI profiles in the patient population?

Reviewer #3:

Remarks to the Author:

Very nice and comprehensive work.

The idea of using PDMS via embedded printing is not the most ideal but the results show that it works as and matches the electro-mimetic performance of cochleae. Supplementary results and video is also supportive of the work.

To improve the quality of the paper, I suggest that the authors also add some information on challenges related to such embedded printing and what future work should be done. What about longer-term performance? Does such a PDMS device can be used multiple times or should be used as a one-time device? Some analysis of mechanical performance from vibration-related fatigue behavior - will that be a factor?

Overall, the results are certainly very exciting.

Response to reviewers

Title: 3D printed biomimetic cochleae and machine learning co-modelling provides clinical informatics for cochlear implant patients

Authors

Iek Man Lei^{1,2}, Chen Jiang^{3,1}, Chon Lok Lei^{4,5}, Simone Rosalie de Rijk³, Yu Chuen Tam⁶, Chloe Swords⁷, Michael P.F. Sutcliffe¹, George G. Malliaras¹, Manohar Bance^{3,*}, Yan Yan Shery Huang^{1,2,*,#}

Affiliations

¹*Department of Engineering, University of Cambridge, United Kingdom*

²*The Nanoscience Centre, University of Cambridge, United Kingdom*

³*Department of Clinical Neurosciences, University of Cambridge, United Kingdom*

⁴*Department of Computer Sciences, University of Oxford, United Kingdom*

⁵*Institute of Translational Medicine, Faculty of Health Sciences, University of Macau, Macau*

⁶*Emmeline Centre for Hearing Implants, Addenbrookes Hospital, Cambridge, United Kingdom*

⁷*Department of Physiology, Development and Neurosciences, Cambridge, United Kingdom*

*Corresponding authors: mlb59@cam.ac.uk, yysh2@cam.ac.uk

#Lead contact: yysh2@cam.ac.uk

REVIEWER COMMENTS

Reviewer #1 (Remarks to the Author):

The paper “A neural network trained by 3D printed biomimetic cochleae provides clinical informatics for cochlear implant patients” presents the development of a neural network model that is informed by measurements in 3D printed biomimetic cochleae with realistic conductivity and anatomy. The model uses 4 geometric features to predict the intracochlear voltage (electric field imaging; EFI). The model has been also used to infer the cochlear geometries based on the measured EFIs.

The research presented in the manuscript is highly innovative and covers a wide range of challenging methods including neural networks, electric field measures in vivo, 3D printed models, finite element method, image analysis through CT and CBCT and a few more. The methods used are highly complex and require a high level of expertise. Unfortunately, the validation of the model predictions in human CI users cannot keep up with this quality. For instance, the 3PNN model has only been validated in 6 subjects with 1 electrode type. Moreover, these 6 subjects had very similar EFI profiles as well as similar geometry descriptors (Fig.5bc).

The option to “infer” the in vivo bone resistivity (which is advertised in the abstract and forms a substantial part of the discussion) is not validated at all.

The reviewer has some general concerns and a list of detailed corrections presented below.

We appreciate the reviewer’s time for his/her careful reading of the manuscript and the constructive comments. We have now addressed the comments, and a detailed point-by-point response to all comments can be found below.

General Comments

1. In general, the methodology presented in the manuscript is innovative and promising and well presented with an exceptional level of detail. The quality of both engineering and scientific work in developing the methodology seems to be very high. Even if it may be logical to think that the spread or width of the EFIs may be related to speech understanding performance, there is no much evidence in the literature for this (see for example: Jürgens et al 2018, PloseOne). In my opinion the third line in the abstract should weakened or look for other alternative main motivations for modeling EFIs.

We appreciate the reviewer’s viewpoint here. Although there are mixed reports on the correlation between spectral resolution and speech intelligibility, there is now a fair amount of evidence from literature supporting the association between current spread and speech comprehension. For example: 1) *Goehring et al, 2020, J. Assoc. Rec. otolaryngol.* shows that simulated current spread in humans degrades speech comprehension; 2) *Srinivasan et al, 2013, Hear. Res.* shows that speech perception in noise is improved with partial tripolar stimulation in subjects as partial tripolar stimulation reduces spread of

excitation; 3) In *Jurgens et al 2018, PLOS One*, they also showed that ‘...*In the model, the wider electrical field spatial spread functions will cause wider modelled neural excitations, resulting in spectrally smeared IRs.*’. Hence, we believe that the undesired spread of excitation should ultimately have a negative impact on speech comprehension, at least beyond a certain limit. Acknowledging the Reviewer’s comment, we have now adjusted the original statement in the revised Abstract (P.1-2).

2. It has been previously reported that EFIs or transimpedance measurements relate to electrode angular depth of electrode insertion (Aebischer et al. 2020, IEEE). Insertion depth depends on different factors including cochlear anatomy (length and shape), surgical procedure or even surgeon. Moreover, it is possible that impedances are affected by electrode-modiolar distance (e.g. Saunders et al 2002, Ear Hear). In your neural network and in your models you did not studied the effect of different electrode locations for the same cochlea on the EFI. In other words, two subjects having the same exact cochlea anatomy may receive the cochlear implant electrode contacts in different locations which would lead to different EFIs, however based on your model the same EFI would be predicted for both subjects as only cochlear anatomical features are used.

We thank the reviewer for this insightful comment.

a. Overall comment

First, it is important to emphasize that 3PNN gives a **statistical** prediction, with **the most likely** outcomes presented by **the clustering of prediction points**. Using the example given by the Reviewer, indeed two subjects having the same cochlea anatomy could have different EFIs due to cochlear implant electrode contacts in slightly different locations. But if we were to acquire EFIs on many pseudo-patients having the same cochlea anatomy, a “most probable” EFI will result. This is because shifts in electrode placement longitudinally or laterally along the cochlear lumen are attributed to extrinsic statistical variability (e.g. surgical practices, etc), which are important to determine individual cases, but are considered as secondary factors. In comparison, the cochlear anatomy is the dominating, intrinsic factor that governs the ‘bounds’ where a CI electrode array can be located.

Secondly, the stimulating electrode positions and the recording electrode positions are the underlying inputs in 3PNN. Patient-specific values of the electrode positions are not required in 3PNN as an assumption that the electrode positions follow the implant specification is taken for all predictions. Supplementary Table 2 provides details of the input values of the electrode positions of different implant types used in 3PNN, as stated in their specifications.

With the above principle in mind, we have now further provided information regarding the effect of electrode locations in our 3D printed models on EFIs and the capability of 3PNN (i.e. 3D printing and neural network co-modelling) in predicting off-stimulation EFIs of different electrode types that have different electrode locations, see below.

b. Comment on electrode-to-modiolus distances

In Fig.4c(i), we demonstrated that the electrode-to-modiolus distances of CI electrodes in human cochleae (replotted from *Davis et al, 2016, Otol. Neurotol.* for Advanced Bionics® HiFocus™ 1J electrode) are statistically close to the electrode-to-spiral centre distance of CI electrodes in our 3D printed biomimetic cochleae. The interquartile range (IQR) of the electrode-to-modiolus distance in human cochleae is 1.2 – 0.9 mm, and the IQR of the 3D printed cochleae is 1.1 – 0.8 mm. This confirms that the cochlear anatomy governs the ‘bounds’ to which a CI electrode array can be located.

c. EFIs measured with different CI electrode types in same 3D printed models

We have now measured EFIs in the same 3D printed models using different types of CI electrode arrays and examined their effects. Supplementary Fig.10a compares the off-stimulation EFI measurements of HiFocus™ 1J electrode array (CI^{1J}) and HiRes™ Ultra HiFocus™ SlimJ electrode (CI^{SlimJ}) ($n = 6$), and the off-stimulation EFI between CI^{1J} and Cochlear Corporation Nucleus® CI522 slim straight electrode (CI522) ($n = 9$). It is important to note that CI^{1J}, CI^{SlimJ} and CI522 were manufactured by different CI companies, and have different electrode spacings (Supplementary Table 2). The 3D printed models here were randomly selected and exhibited different model descriptors (Supplementary Fig.10c). The results show that the off-stimulation EFIs measured in the same model by the different CI electrode types have very similar overall shape and trend, providing confidence that 3PNN can be broadly implemented for different CI electrode arrays and EFIs are predominantly governed by the cochlear electroanatomy.

d. 3PNN predictions for different electrode arrays

To confirm the applicability of 3PNN beyond the CI^{1J} electrode array which is used to generate the training dataset, we applied 3PNN to predict off-diagonal EFI (for up to 18.5 mm) of CI^{SlimJ} and CI522 electrode arrays. Supplementary Fig.10b shows the predicted off-stimulation EFIs of CI^{SlimJ} ($n = 6$) and CI522 ($n = 9$) up to 18.5 mm along the cochlear lumen. The predictions were carried out with their corresponding electrode positions and electroanatomical descriptors as the inputs (Supplementary Table 2). The prediction accuracy of CI^{SlimJ} and CI522 off-stimulation EFIs complies with the prediction accuracy of CI^{1J} EFIs, with ~10% MAPE. Therefore, 3PNN has the capacity to predict EFIs of other electrode types.

e. Unpredictable factors

We agree that there are several unpredictable factors (i.e. surgical variation) that will lead to a change in the insertion depth and affect the EFIs. Thus, in our revised manuscript, we emphasise that 3PNN aims to forecast the most likely EFI outcomes and assumed the electrode positions follow the implant specification in all patient predictions (*P.13, Lines 287-289; P22, Line 471; P31, Lines 694-696*).

To investigate how much the surgical variation in the insertion depth affects EFIs, we evaluated the change in EFIs when there is a ± 2 mm difference in the insertion depth in our 3D printed models, as shown in Supplementary Fig.19b. The median absolute

percentage error (MAPE) between the off-stimulation EFIs is $\sim 8\%$ when the insertion depth was changed by ± 2 mm ($n = 18$). This has been pointed out in the revised manuscript (P.21, Lines 449-454).

CI electrode array	Electrode spacing (mm)	Electrode positions along the CI (mm)
Advanced Bionics® HiFocus™ SlimJ electrode (CI ^{SlimJ})	1.3	[3, 4.3, 5.6, ..., 17.3, 18.6]
Advanced Bionics® HiFocus™ 1J electrode (CI ^{1J})	1.1	[2, 3.1, 4.2, ..., 17.4, 18.5]
Cochlear™ Nucleus® slim straight electrode (CI622)	0.9	[3.85, 4.75, 5.65, ..., 17.35, 18.25]
Cochlear™ Nucleus® slim straight electrode (CI522)	0.9	[3.85, 4.75, 5.65, ..., 17.35, 18.25]

Supplementary Table 2: Input values of the stimulating and the recording electrode positions of different electrode types used in 3PNN. (SI, P.21)

Fig.4c(i): The electrode-to-spiral centre distance measured for the CI electrodes in the biomimetic cochleae, compared to that measured in the human cochleae with the same CI electrode type. (P.10)

Experimental off-stimulation EFIs acquired by different CIs in biomimetic cochleae

3PNN predictions of off-stimulation EFIs of different CIs

Supplementary Fig.10: Applicability of 3PNN on different electrode types. a, Experimental off-stimulation EFIs or transimpedance matrices acquired by either CI^{1J} , CI^{SlimJ} or CI522 in same biomimetic cochlea samples. b, Accuracy of 3PNN in predicting (i) CI522 transimpedance matrices and (ii) CI^{SlimJ} EFIs. c, Specifications of the samples tested here. (SI, P.22)

Supplementary Fig. 19b: (i), Boxplot summarised the MAPEs of the experimental CI^{IJ} EFIs acquired when there was a ± 2 mm variation in the electrode insertion depth in our 3D printed models ($n = 18$). **(ii)**, Comparisons of the EFIs acquired with typical insertion depth and the EFIs subject to ± 2 mm insertion depth variation. The values at the upper right indicate the MAPE between the two EFI profiles. (SI, P.31)

3. Include supplementary table with specs of the 82 printed biomimetic cochlea models
The input parameters of the 82 printed cochlear models are now summarised in Supplementary Table 3 (SI, P.36).
4. Why do you use different threshold criteria for the backward 3PNN model? E.g. 8% MAPE (line 652), 6% MAPE (line 660), 10% MAPE (line 668) etc.
In the inverse 3PNN model, we employed the ABC-SMC (Approximate Bayesian Computation-Sequential Monte Carlo) algorithm that uses a sequence of decreasing MAPE threshold $[\varepsilon_0 > \varepsilon_1 > \varepsilon_2 > \varepsilon_3 > \dots > \varepsilon_f]$ to converge towards the optimal approximate posterior distribution through a number of intermediate posterior distributions. ε_f is the optimal (final) MAPE threshold, which is the smallest achievable MAPE in each patient prediction. In the predictions of the four geometric descriptors, ε_f was found by running the inverse programme with a threshold sequence from 20% to 2% in increments of 0.5%, and ε_f is the smallest MAPE value that the programme can reach from the previous threshold level within two hours. The smallest achievable value depends on the patients' EFIs, therefore different values of MAPE threshold were used.

We have added additional detail about the method of finding the optimal MAPE threshold (ε_f) used in the inverse predictions in the Methods section (P.30, Lines 684-688). A table summarising the optimal MAPE thresholds (ε_f) used in the predictions is added in Supplementary Information (Supplementary Table 4, SI, P38).

5. Not clear how the impedance recordings were conducted. According to Vanpoucke et al (2004), IEEE, the measurement electronics of the AB device may take samples at a rate of 56 kHz. How were you able to sample at 83 kHz? In clinical impedance measures only one value during each stimulation pulse is recorded to characterize the so-called "impedance", so that the frequency of stimulation/recording does not play a role. However, in your impedance spectroscopy you used an 83 kHz frequency, did you use here analog stimulation? How does your impedance spectroscopy relate to the EFI measures with the CI?

We thank the reviewer for pointing this out, this was an error. The 83 kHz stated in the original manuscript was the communication frequency with the software. We used a stimulation rate as per standard of Advanced Bionics[®], which is 56 kHz. All results associated with the stimulation frequency in the manuscript, including the electrochemical impedance spectroscopy and resistivity measurements, are now updated using 56 kHz.

- Effect of reference electrodes in the impedance measures. Typically, EFIs are measured using a stimulating electrode and a reference electrode for the stimulation and a recording electrode and a reference electrode for the recording. Which electrodes did you use in your methods and how much were the measurements affected in your model by the fact that you do not have a head in your cochlear printed models?

In all EFI measurements, including the new data measured by CI^{SlimJ} electrode array and CI522 electrode arrays in the revision, the reference electrode is the extracochlear plate ground of the electrode array (known as the ‘case ground’ of CI^{IJ} and CI^{SlimJ}, or the ‘MP2 plate extracochlear electrode’ of CI522). The case ground electrode was located underneath and was in direct contact with the 3D printed cochlear samples during EFI measurements. We have clarified this in the Methods section (P.25, Lines 552-553).

Although the 3D printed cochlear models do not include the entire head configuration, our 3D printed biomimetic cochlear model exhibits a similar impedance property to the cadaveric cochlea in a head, as shown in Fig.2a. This ‘impedance matching’ ability over a wide frequency range is the key for replicating patients’ EFI profiles. The main text (P.6, Line 131) and the Methods section (P.27, Line 597) are now amended to further emphasize that the EIS measurement of the cadaveric cochlea was conducted in a head.

Fig.2a: Bode plot showing the impedance properties of a cadaveric cochlea in a head, and 3D printed cochlear models made of an electro-mimetic bone matrix and a hydrogel. (P.7)

- The limited characteristics or input features used in the 3PNN to predict EFIs cause that the applications mentioned in the discussion sound too optimistic at this stage of the research. Modeling of tissue growth, ossification of the cochlea and different electrode locations should be incorporated into the model to be able to successfully use the 3PNN for the mentioned clinical applications.

We have now updated the Discussion section (P.21, Lines 457-469), to state that disease modelling would require future validation and further studies.

- Sometimes you give a value followed by the units and sometimes followed by space and the units. Check format.

We have ensured that the format of the units is the same throughout the manuscript.

Detailed Comments

Abstract

9. Line 20-23 “(...) the model can reconstruct patients’ clinical electric field imaging (EFI) profiles arising from off-stimulation positions with a 90% mean accuracy (n=6).”

Only shown for slimJ users with very similar EFI profiles. It would be interesting to see the accuracy for different electrode types and more varying EFI profiles.

We agree with the reviewer that using more electrode types and more varying EFIs would be advantageous in the validation. Thus, in our revised manuscript, we validated the accuracy of 3PNN with three electrode types and a wide range of clinical dataset.

In total, 31 sets of paired patient CT scans and EFI profiles were used. This clinical dataset encompasses 17 Advanced Bionics® HiFocus™ SlimJ electrode (CI^{SlimJ}), 6 Cochlear™ Nucleus® CI622 slim straight electrode (CI622) and 8 Cochlear™ Nucleus® CI522 slim straight electrode (CI522). Further, we also ensure the population representativeness of our data by comparing to 97 clinical EFI data (Supplementary Fig.12a).

As elucidated in the updated Discussion ‘...Starting with our forward-3PNN, we predicted the patients’ off-stimulation EFI profiles based on the four geometric descriptors measured from their CT scans, while taking the matrix resistivity input as 9.3 kΩcm (the mean reported resistivity of live human skulls obtained during surgery²²⁻²⁶, see Supplementary Fig.2a). Without any model adjustment for the different CI types, 28 out of the 31 EFI reconstructions achieve a MAPE (median absolute percentage error) < 12% (Fig.5b and Supplementary Fig.11), despite of the limited resolution of patients’ cochlear CT scans, and the substitution of the unknown patient cochlear tissue resistivities with the reported mean human skull resistivity. For a selected patient (subject 4^{CI522}) whose EFI profile matches the population ($n = 97$) mean EFI, forward-3PNN was shown to achieve a MAPE = 8.6% for the EFI reconstruction (Fig.5bi and Supplementary Fig.12b-c). We further validated our 3PNN by inversely inferring the distribution of the four cochlear geometric descriptors that could match a patient’s off-stimulation EFI profile with a similarity > 89% (Similarity (%) = 1 – MAPE (%)). Comparing the predicted distributions of the geometric descriptors with the corresponding patient’s features measured from their CT scans, the median MAPE is ≤ 8% (Fig.5c and Supplementary Fig.13). This high statistical prediction accuracy demonstrates the capacity of 3PNN to autonomously predict clinical EFIs for different electrode types without further need to adjust the machine learning model that is trained by the dataset acquired from the CI^{IJ}.’

Supplementary Fig.12: a, The clinical EFIs ($n = 31$) used in 3PNN validation to represent the EFI variation in patient population ($n = 97$). **b**, The mean of patients' EFI profiles (or transimpedance matrix profiles) ($n = 97$) and the EFI of subject 4^{C1522}. **c**, Performance of (i) forward-3PNN and (ii) inverse-3PNN on subject 4^{C1522}. (SI, P.25)

Fig.5: b, Validation of forward-3PNN for predicting patient off-stimulation EFI. (i) Representative off-stimulation EFI predictions of each implant type, as compared to the corresponding clinical patient data; and (ii) boxplots summarised the overall performance of forward-3PNN, with the median MAPE of each CI electrode type indicated on the figure. **c**, Overall performance of inverse-3PNN for inferring the patients' cochlear descriptors for different CI electrode types, with the median MAPE stated for each descriptor. (P.15)

Supplementary Fig.11: Full validation results of forward-3PNN. The boxplot at the lower right summarises the MAPE values in all predictions. A median MAPE of 8.6% is observed. (SI, P.24)

Supplementary Fig.13: Full validation results of inverse-3PNN. (SI, P.26)

10. Line 25-27 “This work (...) directly reveals individual patient’s in vivo cochlear tissue resistivity (0.6 – 15.9kΩcm, n=16) by CI telemetry.”

◇ No, this method is not validated. At most, you can claim that your model is able to "estimate" the resistivity, but it is not "revealed".

We agree with the Reviewer and have now changed the word to ‘inferring’ in the sentence (P.2, Line 28).

Introduction

11. Line 32 “restore electrical functions” typically a CI is used to restore “neural function” rather than “electric function”. Even if neural function is based on electric function, electric seems to be too simple.

We have now revised the sentence (P.2, Line 34).

12. Line 35 Reference 4 is about optogenetics... sounds inappropriate here.

The original reference is now replaced by a new reference (P.2, Line 37) (McRackan et al, 2018, Laryngoscope) – ‘Over 500,000 cochlear implants (CIs) have been implanted

worldwide with this number expected to rise with an aging population and expanding indications.'

13. Line 38 Unfortunately, the CI does not preserve the tonotopy completely. There are large misalignments between the electrode locations and the frequencies associated to the electrodes. Maybe rewrite slightly this sentence. Maybe something like CI try to mimic the tonotopical structure of the cochlea ...

We thank the reviewer's suggestion. We have now rewritten the sentence as follows:
'It also attempts, in broad terms, to reproduce the tonotopic architecture of the cochlea by delivering frequency specific programmed stimulation at localised regions of the cochlear lumen.'
(P.2, Lines 40-42).

14. Line 63 Cochlear implants are typically classified as neural implants rather than electric implants.

This sentence is now revised as follows:

'A major limitation of today's neural prostheses is their imprecise control of the administered stimulus, arising from the intrinsic conductive nature of biological tissues^{8,9}, and particularly of the biological fluids in the inner ear^{5,7}' (P.3, Line 64).

15. Line 67 "auditory nerve fibers" instead of "auditory nerves".

This is now corrected (P.3, Line 68).

Results

16. Line 132 "...the operating frequencies of CIs (i.e. $f=1-100\text{kHz}$)"

◇ should this be 1 Hz – 100 kHz or 1 kHz – 100 kHz?

This should be 1 kHz – 100 kHz, and it is now specified more clearly (P.6, Line 131).

17. Line 148 Here you measure the resistivity matrices at 83 kHz. I do not understand how this 83 kHz frequency relates to the EFI recorded through the cochlear implant. In the cochlear implant, the EFI is typically recorded with single pulses and only one sample for each stimulating pulse is recorded. This requires clarification.

Thanks for pointing this out. As discussed above (comment 5), all the graphs and the main text associated with the stimulation frequency are now updated with the correct stimulation rate (56 kHz).

18. Fig. 4a, b In the subfigures below these subpanels, I was wondering where is the round window placed? And whether different electrode locations (electrode insertion depth and electrode insertion trajectories) are possible for the same cochlea. These different electrode locations would result in different EFIs.

The 3D printed models do not structurally capture the round window features. In our experiment, the electrode array was inserted into the opening of the printed cochlear lumen of the models until the distal marker of the CI was positioned at the opening, as depicted in Supplementary Fig.22. This procedure is similar to the standard surgical procedure, where the distal marker of CI is positioned at the round window. This is now stated in the Methods section (P.25, Lines 549-551).

Though the models do not mimic the anatomical feature of round window, we expect that its effect on EFIs is small because the electrode positions in a patient’s cochlea and in a 3D printed model that has similar geometric descriptors are close to the same, as shown in Supplementary Fig.7b. The 3D printed model and the patient’s cochlea were implanted with the same type of electrode array (CI^{StimJ}). The angular insertion depths of the electrode array in that patient’s cochlea and in the 3D printed model are approximately identical, at $\sim 420^\circ$.

Supplementary Fig.22: Photo demonstrating the insertion of a CI electrode array in a biomimetic cochlea during EFI measurements. (SI, P.37)

Supplementary Fig.7: b, The electrode positions in (i) a patient’s cochlea and (ii) the 3D printed cochlea that has similar geometric descriptors. (iii) Overlap of the patient’s x-ray and the μ -CT image of the 3D printed cochlea to show their similarity in the electrode positions. Scale bar = 2 mm. (SI, P.16)

19. Fig.4d Don’t need the “doubled” panels
We have now revised the graph.

Fig.4d: Comparison of the mean patient EFI profile ($n = 97$) and the EFI profiles obtained from 3D printed models made of hydrogel, PDMS and electro-mimetic bone matrix. (P.10)

20. Fig. 4d Not sure if in these plots the impedance of the stimulating electrode is also plotted. Actually it seems that it is plotted given the large impedance at the location corresponding with the stimulating electrode. If so, why is the electrode impedance of the stimulating electrode shown here if this is not used in the models?

We have now removed the impedance of the stimulating electrodes of patients' EFIs, but we would like to keep the impedance of the stimulating electrodes of our models' EFIs as an indication of the stimulation point on the graph. (P.10)

21. Line 281-283 How do you define the "similarity"?

Similarity is defined using the equation below,

$$\text{Similarity (\%)} = 100\% - \text{MAPE (\%)}$$

, where MAPE is the median absolute percentage error between the off-stimulation EFI profile and the predicted profile. We have now added a statistical method section (P.32) in the Methods to denote the formulae used for calculating MAPE and similarity. The similarity equation is also stated at its first appearance in the main text (P.14, Line 301).

22. Fig.5b The results do not look very patient-specific, maybe because of the measurements that are also very similar. However, it would be nice to compare the performance of the 3PNN predictions against an "average" EFI profile and/or to compare predicted EFIs with EFIs from other subjects.

As discussed in comment 9, we have now included more diverse EFI data from 3 electrode types. In total, 31 paired sets of clinical data were used in our validation.

In addition to this, as suggested, we examined the performance of 3PNN in predicting an 'average' EFI profile. The 'average' EFI profile was calculated from 97 clinical EFI intra-operative profiles (91 profiles provided by Advanced Bionics® and 6 EFIs of CI^{1J} electrode array acquired in our centre). As the implant types of the Advanced Bionics® profiles are not known here, the insertion depth of these 97 EFI profiles was assumed to be equal to the suggested insertion depth of CI^{1J} electrode array. We found that subject 4^{CI522} has the closest match to the 'average' EFI profile, with MAPE ~ 6%, see Supplementary Fig.12b. Therefore, subject 4^{CI522}'s EFI is used to represent the mean population EFI profile. The predictive performance of 3PNN on subject 4^{CI522} is high, with 8.6% MAPE in the forward prediction and < 10% MAPE in the inverse prediction. (P.13, Lines 284-307)

Supplementary Fig.12: b, The mean of patients' EFI profiles (or transimpedance matrix profiles) ($n = 97$), and the EFI of subject 4^{CI522}. **c**, Performance of (i) forward-3PNN and (ii) inverse-3PNN on subject 4^{CI522}. (SI, P.25)

23. Fig.5c i) Color-coding is not visible

ii) Are the huge error bars for CT-measured features correct?

Fig.5c is now revised to a new graph. The full validation results of inverse-3PNN is now moved to Supplementary Fig.13. (SI, P.26)

The huge error bars of the CT-measured features arise from the low resolution of patient's clinical CT scans. We approximate the error bars of the CT-measured features as half pixel size of the patient's CT scan. In the response to comment 41, we also performed a sensitivity analysis to evaluate the effect of the uncertainty in the CT measurements on the EFI predictions by 3PNN.

24. Line 304 Predictions are based on data from 6 subjects, where is the 1000 predicted set obtained from?

As stated in the Methods section (P.31, Lines 688-691), the ABC-SMC algorithm used in the inverse predictions estimates a posterior distribution of the model descriptors that satisfies the MAPE threshold defined by the user (i.e. a statistical approach). As the predicted distribution of the model descriptors does not have a closed-form expression, we approximated the distribution by drawing 1000 samples from the predicted distribution. Therefore, the range of each inverse prediction is described by the 1000 sets of the model descriptors.

25. Line 315 Here you present Equation 1 which defines a function to fit the EFIs. It is however well known that the EFIs do not go to zero and have a sometimes a constant ground or base level which needs to be taken into account to fit properly the potential

function shown in Equation (1). Did you observe a constant base level in your EFIs? How did you account for it in the fittings of A and b ?

We thank the reviewer for raising this comment. Yes, we do see a baseline feature in our EFIs, and we agree with the reviewer that the previous fitting function cannot describe the ‘non-zero’ baseline feature of EFIs. Therefore, in our revised manuscript, we employed the following fitting function,

$$|z| = ax^{-b} + C \quad \text{—Eq. 1}$$

, where $|z|$ is the transimpedance magnitude, a and b are the fitting coefficients and C is the baseline of the EFI that is defined as the minimum value of the EFI here. The new fitting function can capture the baseline feature of EFIs as $|z|$ approaches the baseline when $x \rightarrow \infty$.

To ensure that Eq.1 is the best-fit function, we also examined other fitting function forms (Eq.2 and Eq.3) that can potentially describe the baseline feature.

$$|z| = \frac{a}{x} + C \quad \text{—Eq. 2}$$

$$|z| = ae^{-x} + C \quad \text{—Eq. 3}$$

Supplementary Fig.16 evaluates their goodness-of-fit by comparing the MAPE between the actual patients’ EFIs and the expected EFIs obtained from the fitting functions. The graph indicates that Eq.1 outperforms Eq.2 and Eq.3, best-fitting the patients’ EFI data ($n = 75$), with MAPE = 4%. Therefore, Eq.1 is used in our manuscript to parameterise the voltage spread characteristics of EFIs.

In our revised manuscript, we revised the method of evaluating the gradient (dy/dx) of the spread distributions of EFIs (i.e. sharpness of the voltage drop) by calculating $\frac{d|z|}{dx} = -abx^{-b-1}$ at $x = 1\text{mm}$ from the stimulating electrode. Further details can be referred to P.16 - 17 of the revised manuscript.

Supplementary Fig.16: Goodness-of-fit test to evaluate the choice of the fitting forms. (SI, P.29)

26. Line 322 I would define “r” as “the distance between the stimulating and the recording intracochlear electrode” to be more specific
We have now updated the definition of ‘r’ to ‘the distance between the stimulating and the recording intracochlear electrodes’. (*P.16, Lines 344-345*)
27. Lines 322-225 Also in Fig 6. In the EFIs of the apical and basal electrodes it is not possible to fit any potential function towards the apex or toward the base respectively. Did you remove these fits for the analysis of the variables A apex A base, b apex and b base?
Yes, fittings were only performed on spread distributions with at least 4 data points. We have now clarified the method of the fitting analysis in Supplementary Fig.15 (*SI, P.28, Lines 517-518*).
28. Line 337 This may be a typo: neurones \diamond neurons
We have now standardised the spelling. ‘Neurons’ is now used. (*P.17, Line 356*)
29. Fig.6a What do the marker colors stand for (color bar missing)?
Fig.6a is now revised to a new figure (*P.18*). The original color coding is not used.
30. Fig.6d I do not understand the "electrode-distance-distance" plots in (i) as well as the plots in (iii)/(iv)
Fig.6d ii and iii Not clear what are the peaks. Are these the peaks of the second diagonal?
Peaks need to be defined
We apologise for the confusion. We have now improved the graph in the revised manuscript (*Fig.6c, P.18*). The 3D model in c(i) shows the 3D overviews of the locations of the electrode array in the printed lumens that is reconstructed from μ -CT scans. To examine the electrode array locations in the 3D lumens clearly, the locations of the electrodes inside the lumen were viewed via the top view (*Fig.6c(iii)*) and via the side view (*Fig.6c(iv)*). The color-filled regions in Fig.6c(iii) and (iv) outline the boundary of the 3D printed cochlear lumens in the corresponding plane.
- The peak in Fig.6c(ii) is defined as the maximum $|z|$ of the spread distribution at off-stimulation positions. This is now stated in the figure caption (*P.18, Lines 388-389*).

Fig.6: c, Reconstructed 3D μ -CT volumes of the cochlear lumens of the biomimetic cochleae with a CI electrode array inserted (marked green). Scale bar = 2 mm; **(ii)** Off-stimulation EFI profiles of the models with the peaks indicating the maximum $|z|$ of the spread distributions at off-stimulation positions; **(iii)** Top view and **(iv)** side view of the cochlear lumens of the models. (P.18)

31. Fig.7 Color-coding for predicted resistivity is not needed as it is shown at the y-axis. This original Fig.7 is now moved to Supplementary Fig.18 (SI, P.30) and the color-coding is removed.

32. Line 394-422 The “deduction” (I would rather call it “fitting”) of patient-specific bone resistivities lacks validation. This is rather a hypothesis and not a method, this should be pointed out more clearly.

Following the Reviewer’s comment, we have changed the word ‘deducing’ to ‘inferring’ in the sentence (P.19, Line 414), and have now clarified this in the Discussion section (P.21, Lines 457-460). As direct resistivity measurement of cochlear bone in live human remains to be established, we believe that validating this would be intriguing for future research.

Discussion

33. Line 431-432 What about the surgical procedure to insert the cochlear implant, or even the fact that different surgeons insert the same implant differently? An extreme case would be cochleostomy vs round window approach which theoretically should have significant effects on the electrode locations and EFIs which your model do not take into account. It would have been much better the inclusion of different electrode arrays from different manufacturers and having different insertions in the same cochlea to have more variability in measured EFIs.

We agree that the variation in surgical procedure could cause deviation in the insertion depth from the implant specification, and therefore the EFIs. One of the authors is a CI

surgeon, who has used both methods for CI insertion. The distance between the cochleostomy and the round window is usually around 1 – 2 mm in surgeries. Therefore, as discussed in comment 2, we investigated how much the EFIs were changed when varying the insertion depth by ± 2 mm. We observed a 8% median MAPE, and this is considered as a source of uncertainty in our prediction. This is now elucidated in the Discussion (P.21, Lines 449-454).

In addition, we performed an insertion study in a cadaveric cochlea to examine the change in the EFI when the insertion is via a) the round window approach, via b) the cochleostomy approach with the round window left open, and via c) the cochleostomy approach with the round window closed with fascia (see the graph below as reviewer-only information). The results show that EFIs obtained via different insertion approaches are very similar, with alike magnitude and shape, indicating that the effect is insignificant.

As discussed in comment 2, we have now included additional information regarding EFIs measured with different electrode arrays from different manufacturers. The findings in Supplementary Figs.9-13 (SI, P.20-26) demonstrate that the EFIs measured with different electrode types in the 3D printed models have a similar general shape (Supplementary Fig.10a), and 3PNN displays a good accuracy in predicting EFIs of different electrode types.

EFIs obtained in the same cadaveric cochlea when the insertion is via a) the round window approach, via b) the cochleostomy approach with the round window left open, and via c) the cochleostomy approach with the round window closed with fascia.

34. Line 438 Why the electrode locations were not required?

This is because 3PNN takes the assumption that the insertion depth follows the electrode locations suggested by the implant specification unless further specification. The assumption of the electrode location is now emphasised in the Results (P.13, Lines 287-289), Discussion (P.22, Line 471) and Methods (P.31, Lines 694-696) sections.

35. Line 441-442 Two different electrode arrays having different dimensions, including inter-electrode spacings (1.3 mm vs 1.1 mm as acknowledged in line 648), were used for training and testing the algorithm which adds more difficulty to the aim of predicting the EFIs.

We thank the reviewer in suggesting using the same electrode array in training and validating the model. As addressed in comment 2 previously, in Supplementary Figs.9 – 10 (SI, P.20&22), we have showed that EFIs measured with different electrode arrays in same biomimetic cochleae have similar overall shape, and have validated that 3PNN is capable of predicting EFIs of different electrode arrays (CI^{IJ}, CI^{SlimJ} and CI522). These

results give confidence that using different electrode arrays in prediction and training does not complicate the prediction; in addition, 3PNN demonstrates the versatile application for predicting EFIs of different electrode arrays without needing model adjustment.

36. Line 462-465 Your model does not consider ossification of the cochlea, tissue growth across the electrode array and effect of electrode location. In my opinion, the potential application of the 3PNN for the applications here mentioned seems very optimistic at this stage.

We agree that further study is needed to explore the versatility of 3PNN on disease modelling. We have now emphasised this in the Discussion section (*P.21, Lines 464-469*).

Methods

37. Lines 531-534 Are you sure that the acquisition was done at 83 kHz? I thought the AB can record at most at ~50 kHz? Which type of stimulation was used to measure the EFIs, analog stimulation or biphasic pulses? Most manufacturers only use biphasic pulses and clinically only one sample for each stimulation pulse is available. Which reference electrodes were used for stimulation and recording and how do these reference electrodes affect the EFI? How does this relate to the position of the reference electrode in the printed cochlear models?

Thanks for pointing this out. The stimulation rate should be 56 kHz. We have updated all the graphs related to the stimulation rate.

As discussed in comment 6, in all EFI measurements, including the new data measured by CI^{SlimJ} and CI522 electrode arrays in the revision, the reference electrode is the extracochlear plate ground of the electrode array (known as the ‘case ground’ of CI^{IJ} and CI^{SlimJ}, or the ‘MP2 plate extracochlear electrode’ of CI522). This is now stated in the Methods section (*P.25, Lines 551-553*). We did not study the effect of the reference electrodes as the case ground is the only reference electrode available in CI^{IJ} and CI^{SlimJ} electrode arrays. The position of the reference electrode was located underneath and was in direct contact with the 3D printed cochlear samples during EFI measurements to mimic the configuration in patients. In our experiments, we found that the position of the reference electrode does not matter as long as the base thickness of the model is at least ~ 4 mm thick.

38. Line 551 Which spacing d has been used?

The spacing between the two inner electrodes was 8.4 mm in our measurements. This is now specified in the Methods section (*P.26, Line 578*).

39. Line 552 Are you sure that the stimulation rate in one electrode was 83 kHz? Again typically, biphasic pulses are used to record impedances and definitely at much lower rates. Authors should explain in detail how this fast stimulation or recording rate (not clear) was generated with the AB implant.

All results related to the stimulation rate are now corrected using 56 kHz.

40. Line 596-598 “The EFI profiles were collected using the telemetry function of the CI with the AB’s Volta software as part of the intraoperative evaluation at our centre.”

◇ what about postoperative impedance changes?

We did not examine the post-operative impedance changes in this manuscript, since our focus is on evaluating the off-stimulation EFI indicating ‘current spread’. Exploring post-operative EFIs (in particular for on-stimulation) will be of interest in our future work.

41. Line 603 I am a bit concerned about the quality of the CT scans and the variability of the derived anatomical measures. How much did the resolution of the CT scan (0.4x0.4x0.4 mm³) impact the anatomical measures of the cochlea? At least you will have an error of half voxel size. Specially looking at Fig.5 in supplementary material Section II) the resolution of the human CBCT scans looks quite limited.

We understand the reviewer’s concern here. Indeed, the resolution of the patients’ clinical CT scans is not ideal. Therefore, we performed a sensitivity analysis to determine the impact of the uncertainty in the CT measurements on the EFI predictions ($n = 31$). We assume the uncertainty in the CT measurements is equal to half pixel size of the patient CT scan. Supplementary Fig.19a shows the MAPE of the predicted EFIs when one of the input geometric descriptors is subject to \pm uncertainty, in comparison with the predicted EFIs using raw measurements. We found that the impact of the uncertainty in the measurements of basal lumen diameter, cochlear width and cochlear height on the EFI predictions is insignificant, while the uncertainty in the taper ratio might cause an effect on the predictions. However, the overall impact is expected to be low as the median MAPE is $\sim 7\%$. This is now stated in the Discussion section (P.21, Line 449-453) and Supplementary Information (SI, P.31).

Supplementary Fig.19: a, The effect of the uncertainty in the CT measurements of the geometric features on EFI predictions. (SI, P.31)

42. Line 621-621 “(..) a grid search varying the number of hidden layers from 1 to 10 (1, 2, 3, 5, 10) and nodes from 16 to 64 (16, 24, 32, 64) was performed to determine the best performing hyperparameters.”

◇ are these values appropriate? Specially when increasing the number of layers to 3,5,10 it seems that the number of parameters is extremely large in comparison to the small dataset used for training.

The grid search was initially performed on number of hidden layers = [1, 5, 10] and nodes = [16, 24, 32, 64]. As we found that one hidden layer yields the best performance, we then narrowed down the search space, and further examined the performance of our neural network when the number of hidden layers was set to 2 and 3. We found that one

hidden layer still gives us the best performance, hence one hidden layer is used in our final hyperparameter configuration.

Supplementary information

43. Supp. line 94-97 “The rationale for using a 1w/v% NaCl solution is that 1w/v% NaCl solution at ambient temperature has the most similar conductivity to human cerebrospinal fluid, which is 1.79 S/m at body temperature¹⁴, and it is known that the electrical conductivity of cerebrospinal fluid is similar to the conductivity of perilymph.”

◇ provide reference

Perilymph has a similar ionic composition to cerebrospinal fluid as it is in continuity with the cerebrospinal fluid in the subarachnoid space in human cochleae (Nin et al, 2016, Eur J Physiol). Therefore, we used the conductivity of cerebrospinal fluid to approximate the conductivity of perilymph. The sentence is now amended with the reference added (*SI, P.9, Lines 192-198*).

44. Supp. Fig.3b Why such a large difference between Y₂(C) and Y₂(E) ?

Supplementary Fig.3 is now moved to Supplementary Fig.4 (*SI, P.10*). Y₂ of CPE₂ can be affected by bone compositions, cadaver age, anatomical geometries, CI insertion depth, etc. (*Jiang et al, 2020, APL Mater.*; *Swanson, et al, 1972, J. Biomech.*; *Lin et al, 2015, Conf. Proc. IEEE Eng. Med. Biol. Soc.*). Since the 3D printed cochlear sample (sample E) was not fabricated exactly with the same cochlear geometries to cadaver (C), it was reasonable to see some variations.

In cadaveric cochleae, we found that the values of Y₂ varies from 0.5 to 3 nS due to the variation in the cadaveric cochleae (*Jiang et al, 2020, APL Mater.*). This range is of a similar magnitude to the value of Y₂ obtained from sample E, which is ~0.7 nS. Therefore, we believe the difference between Y₂(C) and Y₂(E) is reasonable.

45. Supp. Fig.4a The lowest resistivity reported for live human skulls should be 0.6 kΩcm, not 500 Ωcm – see Supp. Fig.1b and main text (lines 114, 135).

This is now updated (*SI, P.11, Supplementary Fig.5a*).

46. Supp Fig.5 On the small subplots you show the real CT scans. The upper plots show the cochlea after doing image processing (maybe segmentation and some filter for enhancing). How much did these filtering affect the estimation of anatomical dimensions? when enhancing edges and contours of the CBCT scans you modify the grey scale that gives information about the bone. These questions are related to the concerns about the resolution of the scans to estimate the detailed anatomical features of the cochlea.

Supplementary Fig.5 is now moved to Supplementary Fig.6 (*SI, P.14*). We have now remeasured all the patients’ features from their CT scans, and ensured that no filtering effect was used and the greyscale value was not adjusted to avoid errors due to the use of different settings. In the response to comment 41, we studied the sensitivity of the EFI predictions caused by the uncertainty in the CT measurements.

47. Supp. Note 1 i) Line 174: explain ν (Poisson's ratio)

This is now amended (*SI, P.13, Line 284*).

48. ii) For the “Global deformation” part, it is difficult to follow the calculus because the equations are unnecessarily rearranged multiple times.

For instance, use $\gamma=F/AG$ instead of $G=F/A\gamma$. The equation in line 175 is not needed then.

We thank the reviewer for the careful reading. This is now amended (*SI, P.13, Line 282*).

49. iii) Line 182-184: “Therefore, as the global shear strain and local normal strain caused by CI electrode array insertions are insignificant, we expect that the insertion of CI electrode (...) will not impose any significant deformation to the matrix.”

◇ what is your criterion to call these values insignificant? How large would the strain have to be in order to be significant?

Based on our calculation in Supplementary Note 1, the local normal strain and the global shear strain are in an order of magnitude of 10^{-5} (0.001%). As an example, we can imagine that the elongation of the matrix caused by an insertion is only around 0.0001 mm in a cochlear model with a cochlear width of 10 mm. Therefore, we believe that a strain less than 1% can reasonably be considered as insignificant, and the strain of our model is far less than 1%.

50. Supp. Fig.6ab It is very difficult to see the electrode array in these plots

Supplementary Fig.6ab is now moved to Supplementary Fig.1. An inset photo of the electrode array is now added.

Supplementary Fig.1: Finite element models of **a(i)** a simplified spiral cochlea without the intracochlear membrane structures, and **b(ii)** a cochlea with the Reissner's membrane and the Basilar membrane. Scale bar = 2 mm. (*SI, P.4*)

51. Supp. Fig.6e Which boundary condition is applied here? (compare Supp. Fig.7)

Supplementary Fig.6e is now moved to Supplementary Fig.1b. The ground was applied to an infinitely surrounding sphere. This is now specified (*SI, P.6, line 154*).

52. Supp. Fig.7 The influence of the boundary condition is very interesting. What are the consequences? Which boundary condition fits best to the in vivo or the in vitro (3D-printed cochleae) cases? This needs to be addressed in the discussion.

Thanks for the comment. Following the reviewer's suggestion, we performed additional tests to examine which boundary condition fits best with the physical (3D printed cochleae) model cases. In Supplementary Fig.1a(ii), we compare the experimental EFIs of

five 3D printed cochlear models with the simulated EFIs obtained from our COMSOL FEM built with the same model descriptors. These results show that the ground placements affect the EFI result significantly. Although placing the ground at the lumen opening in the COMSOL model (condition I) yields the EFI most matching the physical measurements compared to other types of placements, the resemblance is still not ideal. When performing the same comparative study for a linear geometry, as shown in Supplementary Fig.1a(iv), the MAPE can be $> 180\%$. Although our comparative results can be used to guide the boundary condition choice in FEM, the real implication of these results is to confirm the advantages of using a physical model when the modelling parameters or the physical empirical laws are undetermined and complex. Our 3D printed cochlear models can replicate clinical EFIs with high fidelity, and are robust to changes in measurement configuration. This is now pointed out in the Discussion (P.20, Line 439-441) and Supplementary Information (SI, P.4-6).

Supplementary Fig.1: a(i), Finite element models of a simplified spiral cochlea without the intracochlear membrane structures. Scale bar = 2 mm. **a(ii),** Off-stimulation EFI profiles simulated with the common choices of boundary condition used in literature, in comparison with the experimental results acquired from the corresponding 3D printed cochlear models that have the same electroanatomical model descriptors as the COMSOL models. The location of the ground is indicated in blue (the ground in condition (VI) is applied to an infinitely surrounding

sphere). The values at the upper right indicate the MAPEs between the simulated and the experimental EFIs. a(iii), A finite element model of a linear uncoiled cochlea without the intracochlear membrane structures. Scale bar = 10 mm. a(iv), Off-stimulation EFI profiles simulated with different choices of boundary condition. Full caption description can be found in SI. (SI, P.4-6)

53. Supp. Fig.8 Figure title: “Electrode-to-spiral centre distance”

This figure is now moved to Fig.4c(i) in the main text (P.10) with the correct caption.

54. Supp. Fig. 11c Provide the dimension of the input and output layers.

The original Supplementary Fig.11 is now Supplementary Fig.9 in the revised manuscript. The figure caption is now updated as follows (SI, P.21, Lines 422-428).

‘...In summary, the input layers consist of 7 parameters – basal lumen diameter, taper ratio, cochlear height, cochlear width, matrix resistivity, an array of the stimulating electrode positions and an array of the recording electrode positions; one hidden layer with 32 nodes is used; the output of the NN model is a transimpedance matrix (known as EFI for Advance Bionics® implants, or transimpedance matrix for Cochlear Corporation® implants), of which the dimension equals to the product of the dimension of the recording position array and the dimension of the stimulating position array.’

55. Supp. Tab.1 3rd row, 2nd column: should be “Ld,1turn”

This is now corrected (SI, P.15).

56. Supp. Fig.9 This figure is not referenced at all

The original Supplementary Fig.9 is now Supplementary Fig.21 in the revised manuscript, and it is now referenced in the Methods section (P.24, Lines 521-522).

57. Supp. Fig.13cd i) indicate units (mm) on x-axis in panels c,d (can be confused with electrode number in a,b,e)

ii) it seems that only spread distributions with a minimum of 4 data points have been fitted

The original Supplementary Fig.13 is now Supplementary Fig.15 in the revised manuscript. The unit (mm) is now included in our revised figure (SI, P.28). Yes, only spread distributions with a minimum of 4 data points were fitted. We have added this description in the figure caption (SI, P.28, Lines 517-518).

58. Supp. Fig.14 Flip y-axes?

In our fitting analysis, in order to convert the x-axis to ‘distance between the stimulating and the recording intracochlear electrode along the CI’, we transformed the left part of the curve via flipping the curve over y-axis, $f(x) \rightarrow f(-x)$, and shifted the curve horizontally so that the stimulus positions begins at 0. To better clarify our methodology here, we have now removed this statement in the figure because the conversion to ‘distance between the stimulating and the recording intracochlear electrode’ already indicates the flipping of y axis.

59. Supp. Fig.15 Line 395-396: How did you select the values for the 3D printed model? Some of them are very off-centered. The simplest way would be to use the medians. The original Supplementary Fig.15 is now Supplementary Fig.17 in the revised manuscript. We thank the reviewer for this suggestion. We agree that it is more appropriate to use the median values here. Thus, we fabricated the on-demand patient-specific models using the median values of the predicted model descriptors, as indicated in Supplementary Fig.17.

Supplementary Fig.17: The predicted distribution of the model descriptors of subjects 1^{1J} and 2^{1J}, and the selected parameters for fabricating on-demand patient-specific biomimetic cochleae. (SI, P.29)

Reviewer #2 (Remarks to the Author):

The authors are to be commended for their timely and innovative work. Cochlear models would be very useful for improving cochlear implant electrode design, surgical planning, and basic research to name a few possibilities. To my knowledge, the work is novel and has potential to impact hearing science and patients in the future. Many of my questions focus on the 3D printed models that guide the development of the neural network.

1. Line 92-93: I wasn't clear if your modeling approach resolves the issues concerning boundary conditions. If so, can you point me to where the boundary conditions are discussed for your approach.

We thank the reviewer for his/her valuable comments on our manuscript. Our manuscript focuses on the development of 3PNN – where EFIs acquired from 3D printed cochlear models, along with their physical model descriptors, are used as the input dataset for training a neural network machine learning model.

In our revised manuscript, we investigated which boundary condition in FEM can lead to most similar EFI profiles to the 3D printed cochleae (*in vitro case*), as shown in Supplementary Fig.1a (*SI, P.4-6*) (please also refer to the response to comment 52, reviewer1). Although placing the ground at the lumen opening in FEM (condition I) is found to yield EFI profiles better matching the experimental EFIs of the 3D printed models for the same model descriptors, the model accuracy is low (MAPE between ~25% to ~180% depending on model geometry). This finding highlights the advantage of 3PNN compared to FEM, as FEM has not been able to replicate all underlying physics in the physical system (e.g. conduction at the electrolyte and electrode interface), despite the FEM model replicates all the geometric parameters and bulk materials properties in the 3D printed model (condition VII in Supplementary Fig.1a(ii)). In summary, by generating training dataset using 3D printed models (physical systems), we have bypassed the sensitivity in the choice of boundary conditions and physical laws that were normally faced by FEM. This is pointed out in the introduction (*P.4, Lines 92-93*) and in the discussion (*P.20, Lines 439-441*).

Supplementary Fig.1: a(i), Finite element models of a simplified spiral cochlea without the intracochlear membrane structures. Scale bar = 2 mm. **a(ii)**, Off-stimulation EFI profiles simulated with the common choices of boundary condition used in literature, in comparison with the experimental results acquired from the corresponding 3D printed cochlear models that have the same electroanatomical model descriptors as the COMSOL models. The location of the ground is indicated in blue (the ground in condition (VI) is applied to an infinitely surrounding sphere). The values at the upper right indicate the MAPEs between the simulated and the experimental EFIs. **a(iii)**, A finite element model of a linear uncoiled cochlea without the intracochlear membrane structures. Scale bar = 10 mm. **a(iv)**, Off-stimulation EFI profiles simulated with different choices of boundary condition. Full caption description can be found in SI. (SI, P.4-6)

- Lines 165-170: Great care was exercised in designing the matrix to ensure a CI electrode can be inserted without damage to the electrode. If models are to be used in a patient-specific framework, are the final positions similar to those in patients?

My concern is that later (lines 188-190) we are told that intracochlear structures (e.g., basilar membrane) are not modeled. Does the presence of intracochlear structures not affect electrode insertion? If it does not, please provide evidence.

We acknowledge the reviewer's concern here. In our revised manuscript, we compared the positions of the electrode array in a patient's cochlea and in a 3D printed model that has similar geometric descriptors, as shown in Supplementary Fig.7b. Both the 3D printed

model and the patient's cochlea were implanted with the same type of electrode array (CI^{SlimJ}). Though our 3D printed models do not capture the intracochlear structures, the positions of the electrode array in the patient's cochlea and in the 3D printed model are almost identical, with an angular insertion depth $\sim 420^\circ$. This finding is now added in the Results section (P.11, Lines 233-238).

Supplementary Fig. 7: *b*, The electrode positions in (i) a patient's cochlea and (ii) the 3D printed cochlea that has similar geometric descriptors. (iii) Overlap of the patient's x-ray and the μ -CT image of the 3D printed cochlea to show their similarity in the electrode positions. Scale bar = 2 mm. (SI, P.16)

3. The modeling done in this work is quite complex and involves a number of approximations and assumptions. It would help me if sources of errors in modeling could be tabulated in one place and their effects could be quantified. As an example, cochlear geometry is modeled using a few parameters. How close are these models to that of the patient being modeled?

Global and local metrics as used in the following article would be a nice way to quantify these discrepancies:

Noble JH, Labadie RF, Majdani O, Dawant BM. Automatic segmentation of intracochlear anatomy in conventional CT. IEEE Trans Biomed Eng. 2011;58(9):2625-2632.
doi:10.1109/TBME.2011.2160262

An error map as in Figure 8 of that paper would be useful.

If there are discrepancies between the model geometry and the patient's geometry as revealed by imaging, how do these propagate to errors in EFI simulations?

Similarly, other assumptions and approximations should be considered. A sensitivity analysis would help tremendously.

We thank the reviewer for this insightful comment. We have now provided answers according to the three main points listed below.

a, Comments on the use of 4 geometric parameters in cochlear geometry modelling

1. Reasons for using parametric modelling

The use of the four geometric parameters does not intentionally aim to thoroughly capture the patient's cochlear geometry. There are several reasons why we chose

parametric modelling to model cochlear geometry, rather than modelling the entire 3D surface contour, which is complex and computationally expensive. A key benefit of parametric modelling is the ease of modelling. Simple models with fewer parameters are always preferred over complex models with many parameters due to the simplicity of interpretation, as long as the most relevant parameters are captured in the model. Having more parameters not only increases the demand for experimental data and computational time, but may also increase the noise of the model.

Here, we selected the basal lumen diameter, taper ratio, cochlear width, cochlear height and cochlear resistivity as the modelling parameters because the intracochlear voltage distribution is mainly governed by volume conduction and the angular insertion depth of the electrode array. While the four geometric descriptors control the conduction volume, the cochlear width predominantly controls the angular insertion depth, as shown in Supplementary Fig.7c. The graph also reveals that our 3D printed cochleae display a similar correlation between the angular insertion depth and the cochlear width, compared to the patient's trend. This indicates that the patients' angular insertion depth features can be replicated by 3D printed models that have similar cochlear width.

2. *The discrepancy between the patient's cochlear geometry and the biomimetic cochlear lumen modelled by 4 geometric descriptors*

In Supplementary Fig.7a, the 3D volumes of the patient's cochlea and the biomimetic cochlea that have similar geometric descriptors are overlapped. Despite only 4 parameters are used to model the patient's cochlear geometry, the 3D printed model can capture the approximate shape of the patient's cochlea. A good match is observed between the model's and the patient's geometries from 0° – $\sim 540^\circ$ (~ 1.5 cochlear turn). Though the geometry similarity reduces for insertion beyond ~ 1.5 turn (540°) of the cochlea, we suggest the effect is minor because the electrode array will not reach a depth of 540° in normal scenarios (the mean angular insertion depth in patients is $\sim 420^\circ$, *P.O'Connell et al, 2017, Otol. Neurotol.*). In particular, as mentioned in comment 2 above, when using the same electrode array type in a patient's cochlea and in a 3D printed model that has similar geometric descriptors, the positions of the electrode array are almost identical (Supplementary Fig.7b).

Given the outstanding geometry matching between the patient's cochlea and the printed lumen of the model up to 1.5 turn, we expect that the propagation error in EFI predictions is not significant as the minor differences in surface contour does not affect the configuration of the electrode array (as proven in Supplementary Fig.7b) and the regional volumetric conductance of cochlear lumen.

The above finding is now included in the revised manuscript (*P.11, Lines 233 - 240*).

b, Assumptions and uncertainty in 3PNN

The key assumption taken in all predictions in this study is that the electrode positions of the implant follow the CI specification as 3PNN aims to forecast the statistically, most likely EFI outcomes. This has now been clarified in the Results (*P.13, Lines 287-*

289, Discussion (P.22, Line 471) and Methods (P.31, Lines 694-696) sections, and details of the input values of the electrode positions of different implant types used in this study can be found in Supplementary Table 2 (SI, P.21).

There are two main potential sources of uncertainty in 3PNN, as noted in the Discussion of the revised manuscript (P.21, Lines 449-454). The first is the uncertainty in the clinical CT measurements, which are normally at low resolution quality. We examined its impact on the patients' EFI predictions ($n = 31$) with the assumption that the uncertainty in the clinical CT measurements is equal to half pixel size of the patient clinical CT scan (Supplementary Fig.19a). We found that the impact of the uncertainty in the CT measurements of basal lumen diameter, cochlear width and cochlear height on the EFI predictions is negligible, while the uncertainty in the taper ratio might cause a slight effect on the predictions, with a median MAPE $\sim 7\%$. The second is the uncertainty in the CI electrode insertion depth. In clinical scenarios, the insertion depth in patients could slightly vary from the specification due to different surgical practices. Therefore, we performed a sensitivity analysis to evaluate the change in EFI measurements when there is a ± 2 mm difference in the electrode insertion depth in our 3D printed models (Supplementary Fig.19b). The median MAPE is $\sim 8\%$ when the insertion depth was changed by ± 2 mm ($n = 18$).

Supplementary Fig. 7: **a**, Superimposing the CT volumes of a patient cochlea and the cochlear lumen of a 3D printed biomimetic cochlea that have similar values of the four geometric features. **b**, The electrode positions of the patient and the 3D printed cochlea shown in (a). Scale bar = 2 mm. **c**, The relationship between the angular insertion depth of the CI electrode array and the cochlear width in patients' cochleae ($n = 19$) and in the 3D printed biomimetic cochleae ($n = 8$). (SI, P.16)

Supplementary Fig. 19: *a*, The effect of the uncertainty in the CT measurements of the geometric features on EFI predictions. *b*, The effect of the uncertainty in the electrode insertion depth on EFI measurements. *(i)* MAPEs of the experimental CI^{IJ} EFIs acquired when there was a ± 2 mm variation in the electrode insertion depth in our 3D printed models ($n = 18$). *(ii)* Comparison of the EFIs acquired with typical insertion depth and the EFIs subject to ± 2 mm insertion depth variation. The values at the upper right indicate the MAPE between the two EFI profiles. (SI, P.31)

- It is mentioned that the model can reconstruct patients' EFI profiles arising from off-stimulation positions with a 90% mean accuracy ($n = 6$). From a clinical perspective, what is a sufficient level of accuracy and what is the justification for this level? The accuracy of EFI should be discussed to help the reader appreciate why it is a suitable technique for testing of the model.

Also, does $n = 6$ provide sufficient statistical power in light of the large variability in cochlear anatomy?

a, Comments on the sufficient level of accuracy

To our knowledge, there is no general consensus on the accepted level of accuracy from literature. From our viewpoint, being able to recognise the EFI $\sim 90\%$ mean accuracy would be clinically useful as most clinical tests with a sensitivity of 90% would be considered good tests if the false-positive rate is low. In our revised manuscript (P.13, Lines 284-307), we validated the accuracy of 3PNN (autonomously, with no model adjustment) using varying clinical EFIs acquired from different implant types (17 EFIs from Advanced Bionics[®] HiFocus[™] SlimJ electrode (CI^{SlimJ}), 6 EFIs from Cochlear[™] Nucleus[®] CI622 slim straight electrode (CI622) and 8 EFIs from Cochlear[™] Nucleus[®] CI522 slim straight electrode (CI522)).

Supplementary Fig.11 shows that 28 out of 31 clinical EFI predictions achieve a MAPE < 12% (accuracy > 88%) when substituting the patient's cochlear resistivity with the mean reported resistivity of live human skull (9.3 kΩcm). The median prediction MAPE is 8.6% (boxplot at the lower right, Supplementary Fig.11). The result confirms the high accuracy and the low false-positive rate of 3PNN. To illustrate how much the prediction error is when the MAPE is ~ 8%, the graph below compares the clinical EFI of subject 4^{CI522} and its predicted EFI, which exhibits a MAPE = 8.6%. The prediction closely resembles the clinical EFI.

b, Representativeness of the clinical data used in validation

We have now validated the accuracy of 3PNN using more varying EFIs of different electrode types. In total, as mentioned above, 31 paired sets of patient CTs and EFI data of three different electrode types (CI^{SlimJ}, CI622 and CI522) were used. These 31 clinical data used in validation can represent the EFI variation in patient population, as shown in Supplementary Fig.12a by comparing the 31 clinical data used in validation with 97 clinical EFI data. The validation results shown in Supplementary Fig.11 and Supplementary Fig.13 suggest that both forward-3PNN and inverse-3PNN exhibit a good performance quality, with median MAPE < 9% in forward predictions and median MAPE < 8% in inverse predictions. In addition, a similar performance quality is observed in predicting the patient population mean EFI profile (Supplementary Fig.12c). Clinical profile of subject 4^{CI522} is chosen to represent the patient population mean as the profile is close to the mean EFI profile derived from 97 patients, as shown in Supplementary Fig.12b. The above finding confirms the accuracy of 3PNN in predicting the EFI outcome in normal patient population.

Supplementary Fig.11: Full validation results of forward-3PNN. The boxplot at the lower right summarises the MAPE values of all predictions. (SI, P.24)

Comparison of subject 4^{CI522}'s EFI and the predicted EFI, which exhibits a MAPE of 8.6%.

Supplementary Fig.12: a, The clinical EFIs ($n = 31$) used in 3PNN validation to represent the EFI variation in patient population ($n = 97$). **b**, The mean of patients' EFI profiles (or transimpedance matrix profiles) ($n = 97$) and the EFI of subject 4^{CI522}. **c**, Performance of (i) forward-3PNN and (ii) inverse-3PNN on subject 4^{CI522}. (SI, P.25)

Supplementary Fig.13: Full validation results of inverse-3PNN. (SI, P.26)

- Line 285: It is noted here that “the predictive quality in cochlear height is less satisfactory compared to ...”. How could prediction of cochlear height be improved?

We thank the reviewer for raising this comment. In our opinion, the less satisfactory predictivity for the cochlear height is because the EFIs might be less sensitive to the cochlear height, as the average CI insertion depth in patients is 420° (<1.5 turns) instead of the full height of the cochlea.

- Line 592: It is noted that EFI profiles were chosen randomly from 16 adults. Do these 16 capture the range of variability seen in EFI profiles in the patient population?

As far as we are aware, there is no literature reported the EFI variation in the patient population. To address the reviewer’s concern here, as mentioned in comment 4, we have widely expanded the clinical dataset used in the revised manuscript. In summary, 3PNN was validated with 31 paired sets of clinical EFIs (3 electrode types) and CT scans. The representativeness of these 31 clinical data is ensured by compared the variation in 97 clinical EFIs (91 of them were acquired independently by Advanced Bionics®) in Supplementary Fig.12a (SI, P.25). The mean patient EFI profile used in the revised manuscript was derived from 97 clinical EFIs (Supplementary Fig.12b, SI, P.25), and on-demand patient-specific models were fabricated to reproduce two extreme clinical EFIs (Fig.6b, P.18). We believe the analyses performed in this study is applicable to the variation seen in the patient population.

Reviewer #3 (Remarks to the Author):

Very nice and comprehensive work.

The idea of using PDMS via embedded printing is not the most ideal but the results show that it works as and matches the electro-mimetic performance of cochleae. Supplementary results and video is also supportive of the work.

To improve the quality of the paper, I suggest that the authors also add some information on challenges related to such embedded printing and what future work should be done. What about longer-term performance? Does such a PDMS device can be used multiple times or should be used as a one-time device? Some analysis of mechanical performance from vibration-related fatigue behavior - will that be a factor?

Overall, the results are certainly very exciting.

We thank the reviewer for his/her comments.

a, Challenges in this study

Indeed, there are several challenges associated with 3D printing modelling. The key challenge is material selection. It is challenging to find a material that can be tuned over a wide resistivity range and, at the same time, possesses suitable rheological properties for embedded printing. This difficulty is addressed by creating tuneable interconnected network of sacrificial materials embedded in a PDMS matrix, where the PDMS acts an embedded printing medium at the same time. The second limitation is data generation and interpretation. Without a computational model, large amount of experimental data may be required for recognising trends in a multivariable problem. We thus employed a neural network machine learning model that learned the experimental data behaviour for EFI modelling. As per the reviewer's suggestion, the challenges in this study are discussed in the Results section (*P.7, Lines 150-153*), in the Supplementary Information (*Supplementary Fig.5a, SI, P.11 & Supplementary Note 2, SI, P.17*).

b, Future work

In future investigations, it will be of interest to explore the use of 3PNN in disease modelling, such as abnormal cochlear anatomy and resistivity. Validating the deduction of the patient-specific cochlear resistivity of 3PNN will also be an intriguing area for further research. The above is now included in Discussion (*P.21, Lines 457-469*). Apart from these, future studies can be undertaken to exploit a spatially heterogeneous architecture in the matrix for mimicking the spatially-dependent tissue properties of human cochleae.

c, Long-term performance of the biomimetic cochleae

Regarding the long-term performance of the biomimetic cochleae, as the models are made of PDMS, they can be used multiple times and are long-lasting, as opposed to models made of hydrogels that are lack of stability, mechanical properties and resistivity tuneability. In Supplementary Fig.5c, we examined if the EFI of the same model will

change after a year. The EFIs measured with a time space of one year are almost identical, proving the long-lasting properties of the models.

We believe that vibration during the insertion is unlikely to cause any detrimental effects on the mechanical properties of the model as only very little force is applied to the model during CI insertion (~ 0.004 N, *Majdani et al, 2010, Acta Otolaryngol.*). From our mechanical deformation analysis in Supplementary Note 1 (*SI, P.13*), we found that the local normal strain and the global shear strain are in an order of magnitude of 10^{-5} (0.001%). In Supplementary Fig.5d, we show that the change in the EFI measurements is negligible after 8 insertions. The above finding confirms that the insertion of the implant will not impose any significant deformation to the matrix.

Long-lasting properties and EFI repeatability of the biomimetic cochleae

Supplementary Fig.5: *c*, EFIs measured in the same biomimetic cochlear model before and after a year storage. *d*, EFIs measured in the same cochlear model after multiple CI insertions. (*SI, P.11*)

Reviewers' Comments:

Reviewer #1:

Remarks to the Author:

We are very thankful to the reviewers for addressing all the comments and corrections. The manuscript improved in clarity, especially the inclusion of additional electrode arrays and the good corresponding results clearly strengths this work. Still there are some points that need clarification. One of this points is the description of the method used to record (trans-)impedances with the cochlear implant, which seems insufficient and it could be even incorrect. The authors need to clarify which sampling frequency was used, which stimulation rate was used to measure it (typically only one pulse is used), which stimulation level and which reference electrode was used in each device (cochlear and AB). In the characterization of the impedance, impedance spectroscopy the frequency of 56 kHz is marked. However to this reviewer is not clear the relation between what I believe it is the maximum sampling rate configured in the CI to record a single value of the impedance and the marker of 56 kHz on the spectroscopy.

The second point is the MAPE measure, which for this reviewer it is difficult to interpret. It would be nice to show that MAPE between the EFI of two subjects or the MAPE between the EFI of one subject and the average EFI across subjects, or between the EFI predicted for one subject and the real EFI of another subject. The same analysis could be made for the inverse 3PNN when predicting the cochlear anatomy. It is necessary to understand the range of the MAPE error measure. What is a large error in MAPE and what is a small error in MAPE?

Answer to Previous General Comments

1. Ok
2. a, b. Ok c, d. I do not understand the color-coding of the new Supp. Fig. 10. The legend in a seems to be wrong? Apart from this, it is good that different electrode types have now been used.
- e. Ok
3. Perfect
4. Thank you for the clarification.
5. Not clear how the (tran-s) impedance was measured. See comment above.
6. Ok
7. Ok – sounds more realistic now
8. Ok
9. Supp. Fig. 12a: Nice figure, but “patients” (gray lines) are missing in the legend.
- Supp. Fig. 11 & 13 (full validation of forward & inverse 3PNN) are very good.
10. Ok

Answer to Previous Specific Comments

Introduction

11. Ok
12. Ok
13. Ok
14. Ok
15. Ok

Results

16. Ok
17. I still do not understand the frequency of 56 kHz. This seems to be the maximum sampling rate the device but not its stimulation frequency/rate. Typically a single pulse is used for stimulation and single value is recording within one pulse. The authors should clearly explain the method used to stimulate and record during the impedance measure. Depending on when the time point is sampled within the pulse, the (trans-)impedance can be very different. The voltage rise within the pulse depends on the electrode nerve interface.
18. Supp. Fig. 22: Electrode positions are similar from a top view, but also from a side view? This is related to comment 2 of reviewer2. It is good that the membrane structures have no significant influence on EFI profiles for fixed electrode positions (compare Supp. Fig. 1b), but it is clear that they restrict the placement of the electrode array to the volume corresponding to the scala tympani whereas the array can be inserted into the whole cochlea lumen in the 3D printed models.

Discuss consequences.

19. Good

20. It is not an important point, but the stimulation point is also clearly indicated if the impedance of the stimulating electrode is left out (compare Supp. Fig. 12a,b)

21. Good

22. Inclusion of more electrode types is very good.

The comparison with the "average" EFI profile is not what I meant with the comment. I would like to test the 3PNN predictions versus a "dummy" predictor that always predicts the average EFI profile (for a given electrode type), i.e. without additional knowledge of the patient's individual geometric features. What MAPE do you get with such a "dummy" predictor, and how much better is 3PNN that uses the individual geometric features? This would help to interpret the MAPE values obtained through 3PNN (compare also comment 4 of reviewer2). Same as general comment.

23. Good

24. Ok

25. Good

26. Ok

27. Good

28. Ok

29. Ok

30. Ok

31. Ok – but why did the predicted resistivities for the 1J subjects change? Compare Fig. 7 from the old manuscript and Supp. Fig. 18 from the new manuscript.

32. Good

Discussion

33. Very good

34. Ok

35. Ok

36. Ok

Methods

37. Needs further clarification.

38. Ok

39. 56 kHz was the stimulation rate, or it was the maximum sampling rate of the device?

40. Ok

41. Good

42. Ok

43. Ok

44. Ok

45. Ok

46. Ok

47. Ok

48. Ok

49. Ok

50. Good

51. Ok

52. The influence of the boundary condition on FEM outcomes is very impressive. Need to discuss. Also, it seems that MAPE is not a very good measure of the similarity between EFI profiles: compare first column of Supp. Fig. 1a(ii), MAPE=35% vs MAPE=39%.

53. Ok

54. Good

55. Ok

56. Ok

57. Good

58. The comment corresponded to Supp. Fig. 14 (not 13), which has apparently not been included in the revised manuscript.

59. Good

Reviewer #3:

Remarks to the Author:

The authors made substantial modifications to this article and addressed all issues that I was concerned about.

Reviewer #4:

Remarks to the Author:

The authors have addressed all my comments and improved the manuscript. However there are still some questions remained unanswered.

1- The authors insist that 3PNN is better than FEM since it bypasses the sensitivities in the choice of boundary conditions. This actually raise the question of the accuracy and validity of the 3PNN, as it seems that the multiple assumptions made in the current work are cancelling each other! The significance of these boundary conditions (specially geometry) has been shown by works on spatial arrangement (tonotopy) of cochlea. One clear advantage of the FEM is the possibility of studying individual boundary condition/parameter at a time, without dealing with such combinational effects. This has been done by isolating parameters/boundary condition and studying their effect by sweeping parameters, etc. The authors should either investigate the sensitivity of their model to each boundary condition, or simply clearly state that it is not possible in 3PNN to perform such sensitivity analyses.

2- Thanks for comparing the final insertion in 3D printed model and in one patient (Supp. Fig 7.b). Please comment on having n=1 comparison. Also, it would be great if the authors could show this comparison from other angles as well. In the presented views, it is unclear how far the electrode is placed away from the basilar membrane. For example, Figure 4 b(ii) shows the positioning of the electrode in printed model from a side view, which indicates the trauma caused to basilar membrane due to intrascalar penetration, if such insertion was made in a patient. It would be great to include more of patient-model comparison and provide quantitative analysis of such comparisons.

3- Thanks for adding the sources of uncertainty in the Discussion. It would greatly help the readers if error maps were incorporated, as requested. Also, the authors did not respond to the question regarding the effect of geometrical discrepancies between model and patient's data on EFI profile. Considering the effect of geometry, material properties of solid and fluids being the key factors in electrical conductivity. The authors should clearly specify the sensitivity of their network to each factor, separately. Having EFI profiles matching, does not mean all factors are correct. As requested, the authors should specify/tabulate the source of error in each step and discuss them quantitatively. Supplementary Fig.7 is only one case and the images show some mismatching. Such mismatches should be clearly quantified and discussed.

4- Pietsch's model is limited to cochlear duct length at lateral wall, and does not provide information about the cochlear scalae cross-sections. Authors should provide quantitative comparison of the cochlear scalae cross-sections between their models and human data and discuss the effect of cross-sections on possible mismatch on the EFI measurements.

REVIEWER COMMENTS

Reviewer #1 (Remarks to the Author):

We are very thankful to the reviewers for addressing all the comments and corrections. The manuscript improved in clarity, especially the inclusion of additional electrode arrays and the good corresponding results clearly strengths this work. Still there are some points that need clarification. One of this points is the description of the method used to record (trans-)impedances with the cochlear implant, which seems insufficient and it could be even incorrect. The authors need to clarify which sampling frequency was used, which stimulation rate was used to measure it (typically only one pulse is used), which stimulation level and which reference electrode was used in each device (cochlear and AB). In the characterization of the impedance, impedance spectroscopy the frequency of 56 kHz is marked. However to this reviewer is not clear the relation between what I believe it is the maximum sampling rate configured in the CI to record a single value of the impedance and the marker of 56 kHz on the spectroscopy.

The second point is the MAPE measure, which for this reviewer it is difficult to interpret. It would be nice to show that MAPE between the EFI of two subjects or the MAPE between the EFI of one subject and the average EFI across subjects, or between the EFI predicted for one subject and the real EFI of another subject. The same analysis could be made for the inverse 3PNN when predicting the cochlear anatomy. It is necessary to understand the range of the MAPE error measure. What is a large error in MAPE and what is a small error in MAPE?

We thank the reviewer for his/her valuable comments. Please see below, in blue, our detailed response to the Reviewer's comments. The question numbers in the previous general and specific comments refer to the comment numbers in the first round of revision. We have excluded the comments that have been addressed in the first round of revision.

Comment on the method to record transimpedances

We thank the reviewer for pointing this out again. All the EFI measurements presented in this study (patients' clinical data or 3D printed models) were acquired using the AB volta software of Advanced Bionics® or the Custom Sound® EP 5.1 of Cochlear™ using the default clinical setting. We have now further clarified the stimulation and the recording configurations of the EFI measurements taken by the 'AB volta' software with a senior software engineer from Advanced Bionics®. For the AB volta software, during an EFI measurement, each electrode contact is stimulated one-by-one with a single biphasic stimulation pulse. The pulse width and amplitude for the biphasic pulse stimulation are 36 μ s per phase and 32 μ A, and the maximum recording sampling rate was 56 kHz. For the Custom Sound® EP 5.1 software, the pulse width and amplitude are 25 μ s per phase and 125.3 μ A.

Based on the quoted biphasic pulse (quasi-square wave) of 36 μ s and 25 μ s pulse width, the stimulation rates can be theoretically estimated to have fundamental frequencies of ~14 kHz and 20 kHz, respectively, as described by the Fourier series expansion of a square wave in Eq.1. We have now clarified this information in Method (*P.26, Lines 585 - 590*) and removed the misleading term 'EFI acquisition rate = 56 kHz' throughout the manuscript.

$$f(x) = \frac{4}{\pi} \sum_{n=1,3,5,\dots}^{\infty} \frac{1}{n} \sin(2\pi n f^* x) \quad \text{-- Eq.1}$$

where f^* is the fundamental frequency that is equal to $\frac{1}{2 \times \text{pulse width}} s^{-1} = 14 \text{ kHz}$ (AB Volta) or 20 kHz (Custom sound[®]), $n f^*$ is the n^{th} order harmonic frequency, and x is wave position.

Regarding the matrix impedance characterization based on EIS in Fig.2a (also shown below), we used continuous sinusoid stimulation and recording over the frequency range of 10 Hz – 100 kHz (where 100 kHz is the upper limit of our EIS instrument). As seen in the EIS Bode plot in Fig.2a, the impedance magnitudes of the 3D printed biomimetic cochleae and the cadaveric cochleae were closely matching, which indicates they have similar impedance properties, over a wide frequency range (f) of 10 Hz – 100 kHz. Further, the impedance magnitude reaches a plateau magnitude (i.e. impedance plateau) from ~ 300 Hz. In other words, impedance magnitude measured at high frequencies are relatively frequency-independent. Relevant to the stimulation pulse of 36 μs (AB Volta) and 25 μs (Custom sound[®]) per phase, the associated fundamental frequencies ($f^* \sim 14 \text{ kHz}$ and 20 kHz) lie in the impedance plateau region. Furthermore, the dominant high order harmonics (n) associated with the Fourier series, i.e. for up to $n = 7$ ($14 \times 7 = 98 \text{ kHz}$) and $n = 5$ ($20 \times 5 = 100 \text{ kHz}$) could be captured within the upper 100 kHz limit of the EIS measurement.

Overall, we thank the Reviewer for pointing out the need for a more clear explanation of our experiment here. Instead of saying the matrix impedance was measured at a specific frequency (i.e. 56 kHz), we should have said a ‘representative plateau impedance’ was measured from the EIS. The resistivities of the electro-mimic bone matrices were determined using the plateau impedance magnitudes and their sample sizes. We have now changed the terminology from ‘matrix resistivity (56 kHz)’ to ‘matrix resistivity (plateau value)’.

Fig. 2: Electrical properties of electro-mimetic bone matrices. (P.7)

Comments on the MAPE measure

In response to the Reviewer's comment, we have further clarified the use of MAPE below.

The reason we chose MAPE (median absolute percentage error, Eq.2) as the error measure instead of using other measures such as RMSE (root mean square error, Eq.3) or MAE (mean absolute error, Eq.4) is because of the improved ease of interpreting the values, as MAPE presents the percentage change rather than the absolute change (which is often harder to interpret). In addition, median absolute percentage error is used rather than mean absolute percentage error (Eq.5) because the value of mean can be distorted by outliers, especially in skewed populations. This is now stated in Methods (*P.34, Lines 765 - 767*).

As per the reviewer's suggestion, Supplementary Table.3 shows (1) the MAPE values calculated between the 3PNN prediction (using known patient geometric factors) and the corresponding patient's off-stimulation EFI; and (2) the MAPE values calculated between the patient mean ($n = 97$) and each patient's off-stimulation EFI. The table shows that based on the MAPE value comparison, 3PNN-predictions outperform the 'dummy' patient average. The finding here provides additional confidence in the patient-specific predictability of 3PNN, and is now included in Results (*P.14, Lines 308 - 311*).

Regarding the question about the acceptable error in MAPE, from our viewpoint, being able to recognise the EFI ~ 90% mean accuracy would be clinically useful as most clinical tests with a sensitivity of 90% would be considered good tests if the false-positive rate is low. From our validation results shown in Supplementary Fig.11 (*SI, P.23*), 28 out of 31 EFI predictions achieve a MAPE < 12% (accuracy > 88% and median MAPE = 8.6%). This confirms the accuracy and the low false-positive rate of 3PNN.

$$\text{Median APE} = \text{median of } \left\{ \frac{|\text{Predicted}_1 - \text{Actual}_1|}{\text{Actual}_1}, \frac{|\text{Predicted}_2 - \text{Actual}_2|}{\text{Actual}_2}, \dots, \frac{|\text{Predicted}_i - \text{Actual}_i|}{\text{Actual}_i} \right\} \times 100\% \quad \text{-- Eq. 2}$$

$$\text{RMSE} = \sqrt{\frac{\sum_{i=1}^N (\text{Predicted}_i - \text{Actual}_i)^2}{N}} \quad \text{-- Eq. 3}$$

$$\text{MAE} = \frac{1}{N} \sum_{i=1}^N |\text{Predicted}_i - \text{Actual}_i| \quad \text{-- Eq. 4}$$

$$\text{Mean APE} = \frac{1}{N} \sum_{i=1}^N \frac{|\text{Predicted}_i - \text{Actual}_i|}{\text{Actual}_i} \quad \text{-- Eq.5}$$

	Subjects	MAPE between the 3PNN-predicted EFI and actual subject's EFI (%)	MAPE between the mean patient EFI and the actual subject's EFI (%)
CI ^{Simul}	Subject 1 ^{Simul}	6.6	21.4
	Subject 2 ^{Simul}	7.9	18.9
	Subject 3 ^{Simul}	7.5	13.3
	Subject 4 ^{Simul}	6.8	19.0
	Subject 5 ^{Simul}	7.4	8.1
	Subject 6 ^{Simul}	4.9	17.0

	Subject 7 ^{SlmJ}	9.3	25.2	
	Subject 8 ^{SlmJ}	6.4	23.5	
	Subject 9 ^{SlmJ}	11.8	29.7	
	Subject 10 ^{SlmJ}	5.1	10.8	
	Subject 11 ^{SlmJ}	6.5	17.2	
	Subject 12 ^{SlmJ}	9.7	20.3	
	Subject 13 ^{SlmJ}	9.6	18.9	
	Subject 14 ^{SlmJ}	5.7	11.0	
	Subject 15 ^{SlmJ}	7.4	18.9	
	Subject 16 ^{SlmJ}	8.4	21.4	
	Subject 17 ^{SlmJ}	11.3	11.8	
	CI622	Subject 1 ^{CI622}	11.0	29.0
		Subject 2 ^{CI622}	11.3	23.0
		Subject 3 ^{CI622}	55.8	7.0
		Subject 4 ^{CI622}	10.0	38.2
		Subject 5 ^{CI622}	11.8	40.6
		Subject 6 ^{CI622}	10.3	41.4
CI522	Subject 1 ^{CI522}	7.8	36.2	
	Subject 2 ^{CI522}	10.6	39.2	
	Subject 3 ^{CI522}	7.4	37.5	
	Subject 4 ^{CI522}	8.6	28.2	
	Subject 5 ^{CI522}	10.3	24.5	
	Subject 6 ^{CI522}	22.3	16.2	
	Subject 7 ^{CI522}	60.7	5.0	
	Subject 8 ^{CI522}	6.4	29.9	
	Median	8.6	21.4	
	Interquartile range (IQR)	7.1 – 10.8	16.6 – 29.3	

Supplementary Table 3: Table showing first column, the MAPE value between the 3PNN prediction (using known patient geometric factors) and the corresponding patient' off-stimulation EFI; and second column, the MAPE value calculated from the patient mean (n = 97) and each patient's off-stimulation EFIs. (SI, P.25)

Answer to Previous General Comments

2. a, b. Ok c, d. I do not understand the color-coding of the new Supp. Fig. 10. The legend in a seems to be wrong? Apart from this, it is good that different electrode types have now been used. e. Ok

Thanks for the comment. We have now used a uniform color to represent the EFIs of each electrode type in Supplementary Fig.10a, and used alternating color in Supplementary Fig.10b to represent the EFIs of different samples.

Experimental off-stimulation EFIs acquired by different CIs in biomimetic cochleae

3PNN predictions of off-stimulation EFIs of different CIs

Supplementary Fig.10: Applicability of 3PNN on different electrode types. a, Experimental off-stimulation EFIs or transimpedance matrices acquired by either CI^{I} , CI^{S} or CI522 in same biomimetic cochlea samples. b, Accuracy of 3PNN in predicting (i) CI522 transimpedance matrices and (ii) CI^{S} EFIs. c, Specifications of the samples tested here. (SI, P.22)

5. Not clear how the (tran-s) impedance was measured. See comment above.

Please see our response above and in method (P.26, Lines 584 - 590).

9. Supp. Fig. 12a: Nice figure, but “patients” (gray lines) are missing in the legend.
 Supp. Fig. 11 & 13 (full validation of forward & inverse 3PNN) are very good.

The legend is now updated.

Supplementary Fig.12: a, The clinical EFIs (n = 31) used in 3PNN validation to represent the EFI variation in patient population (n = 97). (SI, P.24)

Answer to Previous Specific Comments

Results

17. I still do not understand the frequency of 56 kHz. This seems to be the maximum sampling rate the device but not its stimulation frequency/rate. Typically a single pulse is used for stimulation and single value is recording within one pulse. The authors should clearly explain the method used to stimulate and record during the impedance measure. Depending on when the time point is sampled within the pulse, the (trans-)impedance can be very different. The voltage rise within the pulse depends on the electrode nerve interface.

The same clinical software (AB Volta) was used to measure the EFI profiles of the 3D printed models, therefore the measurements were taken using the same default setting (i.e. same stimulation pulse width and sampling rate) as the clinical data. Please refer to the response to ‘Comments on the method to record transimpedances’ above and the Method of the revised manuscript (P.26, Lines 584 - 590) for the specification of the default stimulation setting used in the software.

18. Supp. Fig. 22: Electrode positions are similar from a top view, but also from a side view? This is related to comment 2 of reviewer2. It is good that the membrane structures have no significant influence on EFI profiles for fixed electrode positions (compare Supp. Fig. 1b), but it is clear that they restrict the placement of the electrode array to the volume corresponding to the scala tympani whereas the array can be inserted into the whole cochlea lumen in the 3D printed models. Discuss consequences.

Thanks for the comment, we are afraid it is not possible to compare the electrode locations at other angles because only plain x-ray anterior-posterior images were collected in our centre’s routine post-operative patient assessment.

To address the reviewer’s question in an alternative fashion, we performed a FEM analysis to

examine how much the EFI changes when the vertical position (the dimension not captured in our plain x-ray films) of the electrode array in the cochlear lumen is shifted by 0.5 or 1 mm (Supplementary Fig.20b). We found that the effect is negligible with a MAPE just $< 0.5\%$ when the matrix resistivity of the model is set to $9.3 \text{ k}\Omega\text{cm}$ (the mean resistivity of live human skull). The MAPE is increased to $\sim 5\%$ when the vertical position is shifted by 1 mm in an extreme model that has a matrix resistivity of $0.6 \text{ k}\Omega\text{cm}$ (the lowest bound of the resistivity of live human skull). We have now clarified this limitation in our 3D printed models in the Results (P.11, Lines 249 – 251) and Discussion section (P.21, Line 462). Future study can explore the possibility of incorporating the membrane structures in 3D printed cochlear models.

Supplementary Fig.20: b, Using COMSOL simulation, off-stimulation EFI profiles were examined when the z -position (vertical position) of the electrode array is shifted by (a) 0.5 mm, and (b) 1 mm for different matrix resistivities. The value at the upper right indicates the MAPE between the z -shifted and the reference (no shifted) cases. The geometrical features of the COMSOL model are the same as the conditions used in the model without the membrane structures in Supplementary Fig.1b(ii) and the ground was placed at the outer surface of the 8mm radius sphere. (SI, P.34)

22. Inclusion of more electrode types is very good.

The comparison with the “average” EFI profile is not what I meant with the comment. I would like to test the 3PNN predictions versus a “dummy” predictor that always predicts the average EFI profile (for a given electrode type), i.e. without additional knowledge of the patient’s individual geometric features. What MAPE do you get with such a “dummy” predictor, and how much better is 3PNN that uses the individual geometric features? This would help to interpret the MAPE values obtained through 3PNN (compare also comment 4 of reviewer2). Same as general comment.

We thank the Reviewer for pointing out our misunderstanding in your comment in the first round. The response for this comment was addressed in ‘Comments on the MAPE Measure’ above.

31. Ok – but why did the predicted resistivities for the 1J subjects change? Compare Fig. 7 from the old manuscript and Supp. Fig. 18 from the new manuscript.

We thank the Reviewer for pointing this out. The results did not change. There was an error in the graph after we added more clinical data in the first-round revision. Supplementary Fig.19 (old

Supplementary Fig.18) is now updated, and we have now double-checked and ensured that all the patient number labels throughout the manuscript and the SI are consistent.

Supplementary Fig.19: 3PNN estimating patient-specific resistivity of the cochlear tissue. (SI, P.33)

Methods

37. Needs further clarification.

The specification of the stimulation and recording configurations of the EFI measurements was now clarified in the manuscript (P.26, Lines 584 - 590).

39. 56 kHz was the stimulation rate, or it was the maximum sampling rate of the device?

56 kHz is the maximum sampling rate of the device under EFI measurements, and was not the stimulation rate of the CI we used. The stimulation from the AB device is a single biphasic pulse with a pulse width of 36 μ s and pulse amplitude of 32 μ A. In the revised manuscript the information for EFI stimulation and recording was clarified (P.26, Lines 584 - 590).

52. The influence of the boundary condition on FEM outcomes is very impressive. Need to discuss.

Indeed, it was interesting to us that the choice of boundary condition affects the FEM outcomes significantly. This was also illustrated in a previous independent study [1] as well. As our manuscript focuses on developing 3PNN, where FEM was used as a complementary tool, we included the reference in the SI for the readers (SI, P.5, Lines 127 - 129).

[1] Wong, P. et al. Development and validation of a high-fidelity finite-element model of monopolar stimulation in the implanted Guinea pig cochlea. *IEEE Trans. Biomed. Eng.* (2016).

Also, it seems that MAPE is not a very good measure of the similarity between EFI profiles: compare first column of Supp. Fig. 1a(ii), MAPE=35% vs MAPE=39%.

We understand the reviewer’s concern here. This is due to a visual illusion of the graphs caused by 16 off-stimulation spread curves of the two EFIs plotted on the same graph, and different y-axis scale of the two graphs. To show the difference between the two EFIs (condition 1.I vs condition 3.I), below graphs compare the individual stimulation spread curves, which correspond to the individual stimulating electrode, of the condition 1.I EFI (MAPE = 35%) and the condition 3.I EFI (MAPE = 39%). The y-axes of the graphs are set to be in the same scale. From the graphs, we can see that the discrepancy between EFIs in condition 1.I and the discrepancy in condition 3.I are roughly similar, where condition 3.I has a visually slightly larger discrepancy. This is consistent with the information provided from the MAPE values, where the MAPEs of condition 1.I and condition 3.I are of a similar magnitude, and condition 3.I has a slightly larger MAPE than that of condition 1.I. Hence, we believe MAPE is a good and valid measure for this study.

Considering the reviewer’s comment, we have now updated the Supplementary Fig. 1a.ii with the same range and scale in the y-axis.

Figures showing the off-stimulation EFIs simulated from COMSOL using a) condition 1.I and b) condition 3.I, in comparison with their corresponding 3D printed cochlear model that have the same electroanatomical model descriptors as the COMSOL model.

Supplementary Fig. 1a.ii: Off-stimulation EFI profiles simulated with the common choices of boundary condition used in literature, in comparison with the experimental results acquired from the corresponding 3D printed cochlear models that have the same electroanatomical model descriptors as the COMSOL models. (SI, P4)

Reviewer #3 (Remarks to the Author):

The authors made substantial modifications to this article and addressed all issues that I was concerned about.

Reviewer #4 (Remarks to the Author):

The authors have addressed all my comments and improved the manuscript. However there are still some questions remained unanswered.

1- The authors insist that 3PNN is better than FEM since it bypasses the sensitivities in the choice of boundary conditions. This actually raise the question of the accuracy and validity of the 3PNN, as it seems that the multiple assumptions made in the current work are cancelling each other! The significance of these boundary conditions (specially geometry) has been shown by works on spatial arrangement (tonotopy) of cochlea. One clear advantage of the FEM is the possibility of studying individual boundary condition/parameter at a time, without dealing with such combinational effects. This has been done by isolating parameters/boundary condition and studying their effect by sweeping parameters, etc. The authors should either investigate the sensitivity of their model to each boundary condition, or simply clearly state that it is not possible in 3PNN to perform such sensitivity analyses.

Comparison between 3PNN and FEM

We thank the reviewer for the constructive comments. We would like to clarify that we do very much value the advantages of FEM for some uses, and have adopted FEM as an ancillary measure at several points in our study to validate several assumptions undertaken in 3PNN that are not easily examined by 3PNN alone (e.g. how the presence of the intracochlear membranes affects EFIs). Any computational modelling has both advantages and disadvantages. The benefits

of FEM are manifold, which notably include, the possibility of studying individual boundary condition/parameter at a time (as the Reviewer has noted), with a graphical user interface. The 3PNN co-modelling approach proposed in this study provides a complementary, statistical and data-driven approach derived from 3D printed biomimetic cochleae. Key benefits of 3PNN are its effectiveness in deciphering the trend (or sensitivity) in a high-dimensional problem (to be elaborated on in the section below), and its capabilities for inferring the patient's electroanatomical features, and for producing on-demand physical models that generate patients' individual EFIs. In response to the reviewer's comment, we have now included in the Discussion section (P.23, Lines 512 - 513) that by integrating the two methods, would provide new routes to create 'digital twins' of cochleae for cochlear implant patients [see Discussion - '*... Complementary to FEM, the 3D printed biomimetic cochleae offer a robust physical means to replicate the dynamics of ionic conduction and the electron-ion interaction in cochleae with implanted CIs. This is useful as it bypasses the sensitivity in the choice of boundary conditions that are required in FEM (Supplementary Fig. 1a), and it intrinsically captures physical phenomena that could be difficult to replicate fully in FEM. (P.20, Lines 449 - 453)... Adopting machine learning along with parametric descriptions of the cochlear geometry, 3PNN requires only a fraction of the computation time per EFI prediction (estimated 300 times faster) compared to our FEM models (for Intel i5 CPU) (P.21, Lines 474 - 476)... Complemented with FEM, 3PNN could form a building block for future cochlear digital twins for CI testing (P.23, Lines 512 - 513)]*

Comments about the cancelling effect of multiple assumptions

All modelling approaches involve assumptions and simplifications. For example, some of the physical laws in FEM associated with the ionic conduction effects (e.g. ionic conduction at the electrolyte and electrode interface on a molecular level) have yet to be fully established for computational simulation, particularly for dynamically changing conditions, under which polarisation of electrode and electrolyte could take place. In FEM, by model-to-model adjustment of the locations of boundary conditions or setting a voltage offset boundary condition, it was possible to obtain an EFI similar to a patient's. This has been depicted in an independent work by Wong et al. (IEEE, 2016) [1], where a voltage offset on the temporal bone surface was used to yield a close match to the *in vivo* EFI. Further, as shown in Supplementary Fig. 1a of this work, inputting the measured bulk material resistivity and micro-CT geometry of the 3D printed cochleae into the FEM, it still yielded vastly different EFI profiles between simulation and actual data. By arbitrarily adjusting the 'ground position' (i.e. boundary condition), the level of EFI mismatch can be reduced from over 450% to 25%. These findings prove two important points: (1) Many factors, such as the details of material/ solution response to surfaces or current injection etc, are essentially unknown to us; in other words, the associated mechanisms cannot be accounted for explicitly and fully in FEM. (2) Arbitrary adjustment in simulation parameters in FEM, such as boundary conditions (i.e. the ground position) or voltage off-set, could also cause the cancelling of parameters and increase matching of FEM to measured data. Hence, while overall FEM might be possible to construct highly detailed geometries and structures, the physical phenomenon descriptions are likely to still be incomplete.

[1] Wong, P. et al. Development and validation of a high-fidelity finite-element model of monopolar stimulation in the implanted Guinea pig cochlea. *IEEE Trans. Biomed. Eng.* (2016).

How local geometrical discrepancy/ details might have an effect on the EFI modelled

We have now investigated this effect in the response of Comment 3.

Comment on the possibility of performing sensitivity analysis of individual parameters with 3PNN

It is possible to perform sensitivity analysis of how individual model descriptors (or parameters) affect EFIs using 3PNN. Fig.6a(ii) (P.19) is a simple version of such a sensitivity study. It displays the trend of how each model descriptor affects the stimulation spread (quantified by Eq.1 & 2, $|z| = A|x|^{-b} + C$; $|\text{slope}|_{x=1\text{mm}} = \left| \frac{d|z|}{dx} \right|_{x=1\text{mm}} = |-Abx^{-b-1}| = Ab$ at $x = 1\text{mm}$). This was performed by computing 3125 EFIs that are associated with $5 \times 5 \times 5 \times 5 \times 5$ (=3125) combinations of the 5 electroanatomical parameters, to represent the entire modelling space (i.e. 5 evenly spaced values were sampled along the range of each parameter).

In our revised manuscript, we have additionally employed a global sensitivity analysis based on Sobol's method [2] (P.17, Lines 371 – 377; with codes provided in the supplementary dataset). In the Sobol sensitivity analysis, the first order sensitivity indices reflect the independent importance of each parameter (x_1, x_2, \dots) on the model output (y_1, y_2, \dots); the second-order indices indicate the significance caused by the interaction of any two parameters on y_1, y_2, \dots ; and the total-order indices take into account the first-, second- and higher-order indices, such that the indices reflect the variance of y_1, y_2, \dots caused by the parameter solely and together with other parameters. A large value of the sensitivity index (S) implies the greater importance of the associated parameter on the outcome relative to others, and the sum of the first-order indices should be less than or equal to 1.

Sobol's method was used to evaluate the importance of each model descriptor (x_1 to x_5) on the values of the slope of the stimulus spread (defined as $|\text{slope}|_{x=1\text{mm}} = Ab$) and the baseline coefficient, C , denoted in Eqs.1 & 2 in the main text. The first-and the total-order indices (Supplementary Fig.17, SI, P.30) suggest that taper ratio is the most important factor affecting the stimulus spread of EFI; while the matrix resistivity and the cochlear width are the dominant factors affecting C (the baseline). In addition, as implied by the small values of the second-order indices in Supplementary Table 4 (SI, P.31), the variance caused by the interaction of any two model descriptors is less significant.

To further prove the capability of 3PNN for sensitivity analysis on EFI, we use Sobol's method to quantify the overall importance of each model descriptors on the full forms of simulated EFIs (each EFI is a 16x16 matrix) within the model boundary. To note due to the large amount of dataset (twenty 16x16 matrices for first, second and total order results), only the first-order results of the sensitivity analysis on EFI for different model descriptors were provided in Supplementary Table 5 (SI, P.32). The second-order and the total-order indices will be made available from the GitHub repository.

It should be noted that the same number of simulations (~168,000) are required to run in FEM in order to perform a thorough sensitivity analysis of a high dimensional problem (in this case, five parameters with 16x16 matrix). As 3PNN leverages machine learning, it can present a fast and automated prediction of EFIs, where a vast number of predictions can be simulated within a realistic timeframe. As an example, ~0.4 s per simulation was needed in forward-3PNN EFI prediction, versus ~ 2 min per simulation in our simplified FEM using Intel i5 CPU computing power (i.e. 300 times faster). This is particularly important for deciphering trends and performing

sensitivity analysis across each parameter in a high dimensional space, which is the case of our problem.

[2] Sobol, I. M. *Global sensitivity indices for nonlinear mathematical models and their Monte Carlo estimates. Math. Comput. Simul. (2001)*

Supplementary Fig.17: (a) First-order, S_i and (b) total-order, S_T , sensitivity indices obtained from Sobol sensitivity analysis of (i) the coefficient product Ab ($|slope|_{x=1mm}$) toward the apex and the base associated with different stimulating electrodes, and (ii) the coefficient C (the baseline coefficient) associated with Eq.1 and Eq.2 in the main text. BL_d = basal lumen diameter, ρ_{matrix} = matrix resistivity, Tr_a = taper ratio, W_c = cochlear width and h_c = cochlear height. (SI, P.30)

Comment on the possibility of performing sensitivity analysis of boundary conditions with 3PNN

We did not use 3PNN to study the effect of ‘boundary conditions’ normally associated with FEM. This is because the training data were collected from 3D printed (physical) cochlear models with wall thickness > 4 mm where the EFIs are independent of the wall thickness. From our viewpoint, the reason why different boundary conditions were tested in existing FEM studies of cochlear stimulation is because none of these perfectly match the *in vivo* situation, hence different conditions were tested to attempt to match the *in vivo* results. This statement is supported by Wong et al. (IEEE, 2016) [1], which states that ‘...**This is problematic for VCMs (volume**

conduction models) of the cochlea simulating monopolar (MP) stimulation because the return electrode lies outside the physical domain of the model. Existing models deal with this issue by assuming that the end of the auditory nerve is grounded, that the ground is infinitely far away, or that boundary box surfaces are grounded — none of these perfectly match the in vivo situation.

[1] Wong, P. et al. Development and validation of a high-fidelity finite-element model of monopolar stimulation in the implanted Guinea pig cochlea. *IEEE Trans. Biomed. Eng.* (2016).

2- Thanks for comparing the final insertion in 3D printed model and in one patient (Supp. Fig 7.b). Please comment on having n=1 comparison. Also, it would be great if the authors could show this comparison from other angles as well. In the presented views, it is unclear how far the electrode is placed away from the basilar membrane. For example, Figure 4 b(ii) shows the positioning of the electrode in printed model from a side view, which indicates the trauma caused to basilar membrane due to intrascalar penetration, if such insertion was made in a patient. It would be great to include more of patient-model comparison and provide quantitative analysis of such comparisons.

We acknowledge the Reviewer's comment. We have now provided additional imaging examples on the comparison of the CI insertion depths in 3D printed models and in patients' cochleae ($n = 3$ in total). The figure (*Supplementary Fig.7b, SI, P.16*) shows that the insertion depths are satisfactorily matching when their geometrical descriptors are similar. In addition, *Supplementary Fig.7c (SI, P.16)* compares the insertion depth in the 3D printed models and in the patients' cochleae that have different values of the geometrical descriptors. The figure reveals that the 3D printed models result in a similar correlation between the angular insertion depth and the cochlear width, to those seen in patient's ($n > 8$).

For comparing the CI electrode position from other angles, we are afraid that in routine post-operative assessment process conducted in our centre, only plain antero-posterior x-ray images are taken; thus we cannot compare CI positions at other angles. Further, standard clinical CT scans do not have enough resolution to show the position of the basilar membrane, hence it is not possible to assess how far the electrodes are away from the basilar membrane, although assumptions can be made when electrodes clearly pass from the scala tympani to the scala vestibuli. To address the reviewer's question in an alternative manner, we performed a FEM analysis to examine how much the EFI changes when the vertical position of the electrode array in the cochlear lumen is shifted by 0.5 or 1 mm (*Supplementary Fig.20b, SI, P.34*). We found that the effect is negligible with a MAPE just $< 0.5\%$ when the matrix resistivity of the model is set to 9.3 k Ω cm (the mean resistivity of live human skull). The MAPE is increased to $\sim 5\%$ when the vertical position is shifted by 1 mm in an extreme model that has a matrix resistivity of 0.6 k Ω cm (the lowest bound of the resistivity of live human skull).

Regarding the comments on Fig.4b(ii) (*P.10*), there might be some confusion. Fig.4b(ii) shows an exemplar 3D printed cochlear model that is not patient-specific. As stated in the study design (in *lines 184 - 195, P.9 in the main text*), our 3D printed models do not capture the intracochlear membranous structure. Another factor is a limitation of our model already noted in the Discussion (*P.21, Lines 466 - 470*) '*...the 3D printed biomimetic cochleae did not account for the frictional force generated during CI electrode insertions beneath the basilar membrane in human cochleae, which may occasionally cause electrode array buckling or even intracochlear trauma affecting CI performance. We suggest that friction could have attributed to the localised buckling configuration of the CI electrode array captured in the 3D model giving the 'mid-dip' EFI...*'.

Although it would be great to have a patient model that captures the friction and buckling phenomena, typical clinical CT scans do not permit the identification of the membranous structures, and fabricating features in a few μm range is in general challenging with the current 3D printing technology (the reported thicknesses of the Basilar membrane and the Reissner's membrane are $\sim 4\ \mu\text{m}$ and $\sim 2.5\ \mu\text{m}$ in literature [3-4]). Therefore, the incorporation of the membrane structures is beyond the scope of the paper. Future study can explore the possibility of incorporating the membrane structures in 3D printed cochlear models, and coupling computational mechanics in the modelling process.

[3] Frijns, J. H. M., de Snoo, S. L. & Schoonhoven, R. Potential distributions and neural excitation patterns in a rotationally symmetric model of the electrically stimulated cochlea. *Hear. Res.* (1995).

[4] Harada, Y. & Harada, Y. Reissner's membrane. in *Atlas of the Ear* (1983)

Supplementary Fig. 7: a) Comparison of the dimensional discrepancies between patients' cochlear CT and the lumen of the 3D printed biomimetic cochleae that have similar geometrical descriptors. The dimensional discrepancy is encoded with color with a defined tolerance of $\pm 0.3\ \text{mm}$, which is the mean pixel size of the patients' CT scans, using AutoDesk Recap Photo. b) The electrode positions in the patients' cochleae (P) showed in (a) and their corresponding 3D printed models (M). The images on the right show their overlap images. The angular insertion depths are i) $\sim 420^\circ$, ii) $\sim 429^\circ$ and iii) 380° ; c) The relationship between the angular insertion depth of the CI electrode array and the cochlear width in patients' cochleae ($n = 19$) and in the 3D printed biomimetic cochleae ($n = 8$) with different geometric descriptors. (SI, P.16)

Supplementary Fig. 20b: Using COMSOL simulation, off-stimulation EFI profiles were examined when the z -position (vertical position) of the electrode array is shifted by (a) 0.5 mm, and (b) 1 mm for different matrix resistivities. The value at the upper right indicates the MAPE between the z -shifted and the reference (no-shifted) cases. The geometrical features of the COMSOL model are the same as the conditions used in the model without the membrane structures in Supplementary Fig.1b(ii) and the ground was placed at the outer surface of an 8mm radius sphere. (SI, P.34)

3- Thanks for adding the sources of uncertainty in the Discussion. It would greatly help the readers if error maps were incorporated, as requested. Also, the authors did not respond to the question regarding the effect of geometrical discrepancies between model and patient's data on EFI profile. Considering the effect of geometry, material properties of solid and fluids being the key factors in electrical conductivity. The authors should clearly specify the sensitivity of their network to each factor, separately. Having EFI profiles matching, does not mean all factors are correct. As requested, the authors should specify/tabulate the source of error in each step and discuss them quantitatively. Supplementary Fig.7 is only one case and the images show some mismatching. Such mismatches should be clearly quantified and discussed.

Error maps and the effect of geometrical discrepancies between model and patient's data

We have now provided the color-encoded error maps and additional results to examine the dimensional discrepancy between the patients and the models that have similar geometrical descriptors using AutoDesk RaCap Photo (Supplementary Fig.7a, SI, P.16). In the analysis, we defined a tolerance level of ± 0.3 mm, as 0.3 mm is the mean pixel size of patient's CT scans. The figure shows that most region of the 3D volumes up to 1.5 cochlear turn (encapsulates the angular insertion depth in patients) has a dimensional discrepancy within the tolerance level, despite the alignment of the 3D volumes is not optimised here. This is now stated in the Results (P.11, Lines 238 - 242).

We have now also investigated the effect of the geometrical discrepancy on EFIs by comparing the simulated EFIs when the patient's lumen diameter is subject to ± 0.3 mm at different matrix resistivities by (1) 3PNN (by changing the input values of basal lumen diameter and taper ratio) and by (2) FEM (by enlarging and shrinking the 3D volume of the patient's cochlea by ± 0.3 mm). Supplementary Fig.20d (SI, P.34) shows that the resulting MAPEs obtained from both methods range from 2% to 20.8%, depending on the matrix resistivity value. The median MAPE is 4.8% in 3PNN and 8.1% in FEM. This is now pointed out in the discussion (P.21, Lines 463 - 465).

Sensitivity analysis

As mentioned in comment 1, we have now provided a thorough global sensitivity analysis using Sobol's method (*Supplementary Fig.17 and Supplementary Tables 4-5, SI, P.30-32*). The finding is included in the main text (*P.17, Lines 371 – 377*)

Source of errors

Regarding the potential uncertainties in 3PNN, we agree with the reviewer that it is clearer to tabulate the source of uncertainty. The requested information is now provided in Supplementary Table 6 (*SI, P.35*) and discussed in the revised manuscript (*P.21, Lines 458 - 466*).

i) 3PNN						
	Lumen diameter changed by + 0.3 mm			Lumen diameter changed by - 0.3 mm		
	0.6 kΩcm	9.3 kΩcm	26.6 kΩcm	0.6 kΩcm	9.3 kΩcm	26.6 kΩcm
Patient 1	7.9%	3.1%	2.4%	4.9%	4.7%	5.0%
Patient 2	2.7%	4.5%	5.1%	4.3%	11.7%	20.8%
Patient 3	2.0%	3.5%	3.2%	5.9%	9.1%	15.8%
IQR	3.2 – 7.4 %					
Median	4.8%					

ii) FEM						
	Lumen diameter changed by + 0.3 mm			Lumen diameter changed by - 0.3 mm		
	0.6 kΩcm	9.3 kΩcm	26.6 kΩcm	0.6 kΩcm	9.3 kΩcm	26.6 kΩcm
Patient 1	7.9%	8.5%	7.6%	6.1%	10.6%	11.7%
Patient 2	4.9%	8.5%	8.2%	4.0%	10.3%	10.5%
Patient 3	5.5%	6.5%	6.1%	7.4%	10.2%	9.8%
IQR	6.2 – 10.1 %					
Median	8.1%					

Supplementary Table 20d: The MAPE between the simulated EFIs when the lumen diameter is subject to ± 0.3 mm at different matrix resistivities computed by i) 3PNN and ii) FEM. IQR = interquartile range. (*SI, P.34*)

Potential uncertainty in 3PNN	Approximated error on EFI (MAPE)
Absence of the membranous structures in the 3D printed models (Supplementary Fig.1b)	IQR = 2.8 – 5.0% Median = 4% ($n = 5$)
Uncertainty in patients' CT measurements (Supplementary Fig.20a)	BL _d IQR = 2.3 – 4.7%, Median = 2.9% ($n = 31$)
	Tr _a IQR = 4.0 – 16.4%, Median = 6.8% ($n = 31$)
	W _c IQR = 0.4 – 0.5%, Median = 0.5% ($n = 31$)
	h _c IQR = 0.2 – 0.3%, Median = 0.2% ($n = 31$)
Uncertainty in z-position of CI electrode array in cochlear lumen (shifted from 0.5 – 1 mm) (Supplementary Fig.20b)	IQR = 0.3% - 2.8% Median = 0.4% ($n = 6$)
Variation in CI insertion depth due to different surgical practices (± 2 mm) (Supplementary Fig.20c)	IQR = 6.4 – 9.6% Median = 7.9% ($n = 18$)
Geometrical discrepancy between patient cochlea and model's geometry (± 0.3 mm) (Supplementary Fig.20d)	IQR = 3.2 – 7.4% (3PNN) or 6.2 – 10.1% (FEM) Median = 4.8% (3PNN) or 8.1% (FEM) ($n = 18$)

Supplementary Table 6: Potential uncertainties in 3PNN, and their estimated effect on off-stimulation EFIs. IQR = interquartile range. (*SI, P.35*)

4- Pietsch's model is limited to cochlear duct length at lateral wall, and does not provide information about the cochlear scalae cross-sections. Authors should provide quantitative comparison of the cochlear scalae cross-sections between their models and human data and discuss the effect of cross-sections on possible mismatch on the EFI measurements.

This question is related to Q3 above. We are aware that Pietsch's model is limited to cochlear duct length at lateral wall. Similar to our response to Q3 above, Supplementary Fig.7a (*SI. P.16*) shows the dimensional discrepancy in the cross-section of the entire cochlear lumen. Within the CT resolution (0.3 mm mean pixel size) limit, the cross-section between the 3D printed model and the human data match satisfactorily for up to 1.5 turns. Also, as mentioned earlier, in Supplementary Fig.20d (*SI. P.34*), we investigated the effect on EFI when the diameter of the lumen cross-sections is changed by ± 0.3 mm, the discrepancy leads to a median MAPE $< 8.1\%$, depending on the geometry and the matrix conductivity values.

==

Concluding response:

Overall, we thank the Reviewers again for their careful reading of our manuscript. 3PNN does not intentionally aim to capture the full geometrical details of human cochleae. As stated in our study design, as clinical CT scans were used, the inherent CT resolution does not allow for detail construction of cochlear surface contour, nor the inclusion of the membranous structures. Using standard-of-care clinical patient CT scans is practical for clinical translation, as micro-CT scans cannot be performed in living patients. The parametric modelling of the cochlear geometry used in 3PNN, instead of inputting full cochlear CT surface contours, further offers the advantages of ease of interpretation, and reducing data collection and modelling time and costs (as previously noted, 3PNN simulation is about 300 times faster than a simple FEM for predicting patient EFI). Finally, a model is useful if it is fit-for-purpose. 3PNN was demonstrated to be predictive for correlating the off-stimulation EFI and the geometric parameters collected from clinical patient CTs (as shown in Supplementary Fig.11 (*SI, P.23*), where we tested for four implant types in an autonomous fashion without model adjustments, and 28 out of 31 predictions show good accuracy, MAPE $< 12\%$ and median MAPE = 8.6%).

Reviewers' Comments:

Reviewer #1:

Remarks to the Author:

The manuscript has much improved during the review process and should now be accepted - no further comments or questions from my side.

Reviewer #4:

Remarks to the Author:

The authors addressed all issues that I was concerned about.